# One-shot Empirical Privacy Estimation for Federated Learning

**Galen Andrew**[*§]  **Peter Kairouz**[*]  **Sewoong Oh**[*]
**Alina Oprea**[*†]  **H. Brendan McMahan**[*]  **Vinith M. Suriyakumar**[‡]

## Abstract

Privacy estimation techniques for differentially private (DP) algorithms are useful for comparing against analytical bounds, or to empirically measure privacy loss in settings where known analytical bounds are not tight. However, existing privacy auditing techniques usually make strong assumptions on the adversary (e.g., knowledge of intermediate model iterates or the training data distribution), are tailored to specific tasks, model architectures, or DP algorithm, and/or require retraining the model many times (typically on the order of thousands). These shortcomings make deploying such techniques at scale difficult in practice, especially in federated settings where model training can take days or weeks. In this work, we present a novel "one-shot" approach that can systematically address these challenges, allowing efficient auditing or estimation of the privacy loss of a model during the same, single training run used to fit model parameters, and without requiring any *a priori* knowledge about the model architecture, task, or DP training algorithm. We show that our method provides provably correct estimates for the privacy loss under the Gaussian mechanism, and we demonstrate its performance on well-established FL benchmark datasets under several adversarial threat models.

## 1 Introduction

Federated learning (FL) (McMahan et al., 2017; Kairouz et al., 2021b) is a paradigm for training machine learning models on decentralized data. At each round, selected clients contribute model updates to be aggregated by a server, without ever communicating their raw data. FL incorporates *data minimization* principles to reduce the risk of compromising anyone's data: each user's data never leaves their device, the update that is transmitted contains only information necessary to update the model, the update is encrypted in transit, and the update exists only ephemerally before being combined with other clients' updates and then incorporated into the model (Bonawitz et al., 2022). Technologies such as secure aggregation (Bonawitz et al., 2017; Bell et al., 2020) can be applied to ensure that even the central server cannot inspect individual updates, but only their aggregate.

However, these data minimization approaches cannot rule out the possibility that an attacker might learn some private information from the training data by directly interrogating the final model (Carlini et al., 2021; Balle et al., 2022; Haim et al., 2022). To protect against this, *data anonymization* for the model is required. FL can be augmented to satisfy user-level differential privacy (Dwork and Roth, 2014; Abadi et al., 2016; McMahan et al., 2018), the gold-standard for data anonymization. DP can guarantee each user that a powerful attacker – one who knows all other users' data, all details about the algorithm (other than the values of the noise added for DP), and every intermediate model update – still cannot confidently infer the presence of that user in the population, or anything about their data. This guarantee is typically quantified by the parameters $\varepsilon$ and $\delta$, with lower values corresponding to higher privacy (less confidence for the attacker).

DP is often complemented by *empirical privacy estimation* techniques, such as membership inference attacks (Shokri et al., 2017; Yeom et al., 2018; Carlini et al., 2022), which measure the success of an

---

[*]Google
[†]Northeastern University
[‡]MIT
[§]Corresponding author: `galenandrew@google.com`

adversary at distinguishing whether a particular record was part of training or not.[1] Such methods have been used to audit the implementations of DP mechanisms or claims about models trained with DP (Jagielski et al., 2020; Nasr et al., 2021; Zanella-Béguelin et al., 2023; Lu et al., 2022). They are also useful for estimating the privacy loss in cases where a tight analytical upper bound on $\varepsilon$ is unknown, for example when clients are constrained to participate in at most some number of rounds, or when the adversary does not see the full trace of model iterates. However, existing privacy auditing techniques suffer from several major shortcomings. First, they require retraining the model many times (at least thousands) to provide reliable estimates of DP's $\varepsilon$ (Jagielski et al., 2020; Nasr et al., 2021). Second, they often rely on knowledge of the model architecture and/or the underlying dataset (or at least a similar, proxy dataset) for mounting the attack. For example, a common approach is to craft a "canary" training example on which the membership is being tested, which typically requires an adversary to have access to the underlying dataset and knowledge of the domain and model architecture. Finally, such techniques usually grant the adversary unrealistic power, for example (and in particular) the ability to inspect all model iterates during training (Maddock et al., 2022), something which may or may not be reasonable depending on the system release model.

Such assumptions are particularly difficult to satisfy in FL due to the following considerations:

- **Minimal access to the dataset, or even to proxy data.** A primary motivating feature of FL is is that it can make use of on-device data without (any) centralized data collection. In many tasks, on-device data is more representative of real-world user behavior than any available proxy data.

- **Infeasibility of training many times, or even more than one time.** FL training can take days or weeks, and expends resources on client devices. To minimize auditing time and client resource usage, an ideal auditing technique should produce an estimate of privacy during the same, single training run used to optimize model parameters, and without significant overhead from crafting examples or computing additional "fake" training rounds.

- **Lack of task, domain, and model architecture knowledge.** A scalable production FL platform is expected to cater to the needs of many diverse ML applications, from speech to image to language modeling tasks. Therefore, using techniques that require specific knowledge of the task and/or model architecture makes it hard to deploy those techniques at scale in production settings.

In this paper, we design an auditing technique tailored for FL usage with those considerations in mind. We empirically estimate $\varepsilon$ efficiently under user-level DP federated learning by measuring the training algorithm's tendency to memorize arbitrary clients' updates. We insert multiple canary clients in the federated learning protocol with independent random model updates, and design a test statistic based on the cosine angle of each canary update with the final model to test participation of that canary client in the protocol. The intuition behind the approach comes from the elementary result that in a high-dimensional space, isotropically sampled vectors are nearly orthogonal with high probability. So we can think of each canary as estimating the algorithm's tendency to memorize along a dimension of variance that is independent of the other canaries, and of the true model updates.

Our method has several favorable properties. It can be applied during the same, single training run which is used to train the federated model parameters, and therefore does not incur additional performance overhead. Although it does inject some extra noise into the training process, the effect on model quality is negligible, provided model dimensionality and number of clients are reasonably sized. We show that in the tractable case of a single application of the Gaussian mechanism, our method provably recovers the true, analytical $\varepsilon$ in the limit of high dimensionality. We evaluate privacy loss for several adversarial models of interest, for which existing analytical bounds are not tight. In the case when all intermediate updates are observed and the noise is low, our method produces high values of $\varepsilon$, indicating that an attacker could successfully mount a membership inference attack. However, in the common and important case that only the final trained model is released, our $\varepsilon$ estimate is far lower, suggesting that adding a modest amount of noise is sufficient to prevent leakage, as has been observed by practitioners. Our method can also be used to explore how leakage changes as aspects of the training protocol change, for which no tight theoretical analysis is known, for example if we limit client participation. The method we propose is model and dataset agnostic, so it can be easily applied without change to any federated learning task.

---

[1]Some prior work only applies to example-level DP, in which *records* correspond to examples, as opposed to user-level, in which *records* are users. We will describe our approach in terms of user-level DP, but it can be modified to provide example-level DP by using DP-SGD in place of DP-FedAvg.

## 2 BACKGROUND AND RELATED WORK

**Differential privacy.** Differential privacy (DP) (Dwork et al., 2006; Dwork and Roth, 2014) is a rigorous notion of privacy that an algorithm can satisfy. DP algorithms for training ML models include DP-SGD (Abadi et al., 2016), DP-FTRL (Kairouz et al., 2021a), and DP matrix factorization (Denissov et al., 2022; Choquette-Choo et al., 2023). Informally, DP guarantees that a powerful attacker observing the output of the algorithm $A$ trained on one of two *adjacent* datasets (differing by addition or removal of one record), $D$ or $D'$, cannot confidently distinguish the two cases, which is quantified by the privacy parameters $\epsilon$ and $\delta$.

**Definition 2.1. User-level differential privacy.** The training algorithm $A : \mathcal{D} \to \mathcal{R}$ is user-level $(\epsilon, \delta)$ differentially private if for all pairs of datasets $D$ and $D'$ from $\mathcal{D}$ that differ only by addition or removal of the data of one user and all output regions $R \subseteq \mathcal{R}$:

$$\Pr[A(D) \in R] \leq e^\epsilon \Pr[A(D') \in R] + \delta.$$

DP can be interpreted as a hypothesis test with the null hypothesis that $A$ was trained on $D$ and the alternative hypothesis that $A$ was trained on $D'$. False positives (type-I errors) occur when the null hypothesis is true, but is rejected, while false negatives (type-II errors) occur when the alternative hypothesis is true, but is rejected. Kairouz et al. (2015) characterized $(\epsilon, \delta)$-DP in terms of the false positive rate (FPR) and false negative rate (FNR) achievable by an acceptance region. This characterization enables estimating the privacy parameter as:

$$\hat{\epsilon} = \max \left\{ \log \frac{1 - \delta - \text{FPR}}{\text{FNR}}, \log \frac{1 - \delta - \text{FNR}}{\text{FPR}} \right\}. \tag{1}$$

**Private federated fearning.** DP Federated Averaging (DP-FedAvg) (McMahan et al., 2018) is a user-level DP version of the well-known Federated Averaging (FedAvg) algorithm (McMahan et al., 2017) for training ML models in a distributed fashion. In FedAvg, a central server interacts with a set of clients to train a global model iteratively over multiple rounds. In each round, the server sends the current global model to a subset of clients, who train local models using their training data, and send the model updates back to the server. The server aggregates the model updates via the Gaussian mechanism, in which each update is clipped to bound its $\ell_2$ norm before averaging and adding Gaussian noise proportional to the clipping norm sufficient to mask the influence of individual users, and incorporates the aggregate update into the global model. DP-FedAvg can rely on privacy amplification from the sampling of clients at each round, but more sophisticated methods can handle arbitrary participation patterns (Kairouz et al., 2021a; Choquette-Choo et al., 2023).

**Privacy auditing.** Privacy auditing (Ding et al., 2018; Liu and Oh, 2019; Gilbert and McMillan, 2018; Jagielski et al., 2020) is a set of techniques for empirically auditing the privacy leakage of an algorithm. The main technique used for privacy auditing is mounting a membership inference attack (Shokri et al., 2017; Yeom et al., 2018; Carlini et al., 2022) and translating the success of the adversary into an $\varepsilon$ estimate using Equation (1) directly.

Most privacy auditing techniques (Jagielski et al., 2020; Nasr et al., 2021; Lu et al., 2022; Zanella-Béguelin et al., 2023) have been designed for centralized settings, with the exception of CAN-IFE (Maddock et al., 2022), suitable for privacy auditing of federated learning deployments. CANIFE operates under a strong adversarial model, assuming knowledge of all intermediary model updates, as well as local model updates sent by a subset of clients in each round of training. CANIFE crafts data poisoning canaries adaptively, with the goal of generating model updates orthogonal to updates sent by other clients in each round. We argue that when the model dimensionality is sufficiently high, such crafting is unnecessary, since a randomly chosen canary update with already be essentially orthogonal to the true updates with high probability. CANIFE also computes a *per-round* privacy measure, which it extrapolates into a measure for the entire training run by estimating an equivalent per-round noise $\hat{\sigma}_r$, and then composing the RDP of the repeated Poisson subsampled Gaussian mechanism. However, in practice FL systems do not use Poisson subsampling due to the infeasibility of sampling clients i.i.d. at each round. Our method flexibly estimates the privacy loss in the context of arbitrary participation patterns, for example passing over the data in epochs, or the difficult-to-characterize *de facto* pattern of participation in a deployed system, which may include techniques intended to amplify privacy such as limits on client participation within temporal periods such as one day.

We empirically compare our approach with CANIFE in Appendix G and discuss the assumptions on the auditor's knowledge and capabilities for all recent approaches (including ours) in Appendix H.

## 3 ONE-SHOT PRIVACY ESTIMATION FOR THE GAUSSIAN MECHANISM

As a warm-up, we start by considering the problem of estimating the privacy of the Gaussian mechanism, the fundamental building block of DP-SGD and DP-FedAvg. To be precise, given $D = (x_1, \cdots, x_m)$, with $\|x_j\| \leq 1$ for all $j \in [m]$, the output of the Gaussian vector sum query is $A(D) = \bar{x} + \sigma Z$, where $\bar{x} = \sum_j x_j$ and $Z \sim \mathcal{N}(0, I)$. Without loss of generality, we can consider a neighboring dataset $D'$ with an additional vector $x_0$ with $\|x_0\| \leq 1$. Thus, $A(D) \sim \mathcal{N}(\bar{x}, \sigma^2 I)$ and $A(D') \sim \mathcal{N}(\bar{x} + x_0, \sigma^2 I)$. For the purpose of computing the DP guarantees, this mechanism is equivalent to analyzing $A(D) \sim \mathcal{N}(0, \sigma^2)$ and $A(D') \sim \mathcal{N}(1, \sigma^2)$ due to spherical symmetry.

The naive approach for estimating the $\varepsilon$ of an implementation of the Gaussian mechanism would run it many times (say 1000 times), with half of the runs on $D$ and the other half on $D'$. Then the outputs of these runs are shuffled and given to an "attacker" who attempts to determine for each output whether it was computed from $D$ or $D'$. Finally, the performance of the attacker is quantified in terms of FPR/FNR, and Eq. (1) is used to obtain an estimate of the mechanism's $\varepsilon$ at a target $\delta$.

We now present a provably correct approach for estimating $\varepsilon$ by running the mechanism *only once*. The key idea is to augment the original dataset with $k$ canary vectors $c_i$ for $i \in [k]$, sampled i.i.d. uniformly at random from the unit sphere $\mathbb{S}^{d-1} = \{x \in \mathbb{R}^d : \|x\| = 1\}$, obtaining $D = (x_1, \cdots, x_m, c_1, \cdots, c_k)$. We consider $k$ neighboring datasets, each excluding one of the canaries, i.e., $D'_i = D \setminus \{c_i\}$ for $i \in [k]$. We run the Gaussian mechanism once on $D$ and use its output to compute $k$ test statistics $\{g_i\}_{i \in [k]}$, the cosine of the angles between the output and each one of the $k$ canary vectors. We use these $k$ cosines to estimate the distribution of test statistic on $D$ by computing the sample mean $\hat{\mu} = \frac{1}{k} \sum_{i=1}^k g_i$ and sample variance $\hat{\sigma}^2 = \frac{1}{k} \sum_{i=1}^k (g_i - \hat{\mu})^2$ and fitting a Gaussian $\mathcal{N}(\hat{\mu}, \hat{\sigma}^2)$. To estimate the distribution of the test statistic on $D'$, it would seem we need to run the mechanism on each $D'_i$ and compute the cosine of the angle between the output vector and $c_i$. This is where our choice of $(i)$ independent isotropically distributed canaries and $(ii)$ cosine angles as our test statistic are particularly useful. The distribution of the cosine of the angle between an isotropically distributed *unobserved* canary and the mechanism output (or any independent vector) can be described in a closed form; there is no need to approximate this distribution with samples. We will show in Propositions 3.1 and 3.2 that this distribution can be well approximated by $\mathcal{N}(0, 1/d)$. Now that we have models of the distribution of the test statistic on $D$ and $D'$, we estimate the $\varepsilon$ of the mechanism using the method given in Appendix A which allows us to compute the exact $\varepsilon$ when the null and alternate hypotheses are arbitrary Gaussians. Our approach is summarized in Algorithm 1.

---

**Algorithm 1** One-shot privacy estimation for Gaussian mechanism.

1: **Input:** Vectors $x_1, \cdots, x_m$ with $\|x_j\| \leq 1$,
    DP noise variance $\sigma^2$, and target $\delta$
2: $\rho \leftarrow \sum_{j \in [m]} x_j$
3: **for** $i \in [k]$ **do**
4:     Draw $c_i$ i.i.d. from $\mathbb{S}^{d-1}$
5:     $\rho \leftarrow \rho + c_i$
6: Release $\rho \leftarrow \rho + \mathcal{N}(0, \sigma^2 I)$
7: **for** $i \in [k]$ **do**
8:     $g_i \leftarrow \langle c_i, \rho \rangle / \|\rho\|$
9: $\hat{\mu}, \hat{\sigma} \leftarrow \mathbf{mean}(\{g_i\}), \mathbf{std}(\{g_i\})$
10: $\hat{\varepsilon} \leftarrow \varepsilon(\mathcal{N}(0, 1/d) \,\|\, \mathcal{N}(\hat{\mu}, \hat{\sigma}^2); \delta)$

---

**Proposition 3.1.** *For $d \in \mathbb{N}, d \geq 2$, let $c$ be sampled uniformly from $\mathbb{S}^{d-1}$, and let $\tau_d = \langle c, v \rangle / \|v\| \in [-1, 1]$ be the cosine similarity between $c$ and some arbitrary independent nonzero vector $v$. Then, the probability density function of $\tau_d$ is*

$$f_d(t) = \frac{\Gamma(\frac{d}{2})}{\Gamma(\frac{d-1}{2})\sqrt{\pi}} (1 - t^2)^{\frac{d-3}{2}}.$$

(All proofs are in Appendix B.) The indefinite integral of $f$ can be expressed via the hypergeometric function, or approximated numerically, but we use the fact that the distribution is asymptotically Gaussian with mean zero and variance $1/d$ as shown in the following proposition.

**Proposition 3.2.** *Let $\tau_d$ be the random variable of Proposition 3.1 in $\mathbb{R}^d$. Then $\tau_d \sqrt{d}$ converges in distribution to $\mathcal{N}(0,1)$ as $d \to \infty$, i.e., $\forall \lambda \in \mathbb{R}$, $\lim_{d \to \infty} \mathbb{P}(\tau_d \leq \lambda/\sqrt{d}) = \mathbb{P}_{Z \sim \mathcal{N}(0,1)}(Z \leq \lambda)$.*

For $d \geq 1000$ or so the discrepancy is already extremely small: we simulated 1M samples from the correct cosine distribution with $d = 1000$, and the strong Anderson-Darling test failed to reject the null hypothesis of Gaussianity at even the low significance level of 0.15.[2] Therefore, we propose to use $\mathcal{N}(0, 1/d)$ as the null hypothesis distribution when dimensionality is high enough.

Finally, one might ask: what justifies fitting a Gaussian $\mathcal{N}(\hat{\mu}, \hat{\sigma}^2)$ to the samples for the alternate distribution? While it appears empirically that the distribution of the cosine angles under the alternate distribution is asymptotically Gaussian, there are some subtleties in proving this, as we discuss in Appendix C. But the important thing is that under this choice, as $d$ becomes large, Algorithm 1 outputs an estimate $\hat{\varepsilon}$ that is asymptotically equal to the correct $\varepsilon$ of the mechanism we are auditing.

**Theorem 3.3.** *For $d \in \mathbb{N}$. For some $m$ (which may grow with $d$), let $x_{d1} \ldots x_{dm}$ such that $\|X_d\| = o\left(\sqrt{d}\right)$ where $X_d = \sum_{j=1}^m x_{dj}$. Let $k = o(d)$, but $k = \omega(1)$, and for $i \in [k]$, let $c_{di}$ be sampled i.i.d. uniformly from $\mathbb{S}^{d-1}$. Let $Z_d \sim \mathcal{N}(0; I_d)$. For $\sigma \in \mathbb{R}^+$, define the mechanism result $\rho_d = X_d + \sum_{i=1}^k c_{di} + \sigma Z_d$, and the cosine values $g_{di} = \frac{\langle c_{di}, \rho_d \rangle}{\|\rho_d\|}$. Write the empirical mean of the cosines $\hat{\mu}_d = \frac{1}{k} \sum_{i=1}^k g_{di}$, and the empirical variance $\hat{\sigma}_d^2 = \frac{1}{k} \sum_{i=1}^k \left(g_{di} - \hat{\mu}_d\right)^2$. Let $\hat{\varepsilon}_d = \varepsilon(\mathcal{N}(0, 1/d) \,\|\, \mathcal{N}(\hat{\mu}_d, \hat{\sigma}_d^2); \delta)$. Then $\hat{\varepsilon}_d$ converges in probability to the constant $\varepsilon(\mathcal{N}(0, \sigma^2) \| \mathcal{N}(1, \sigma^2); \delta)$.*

Running the algorithm with moderate values of $d$ and $k$ already yields a close approximation. We simulated the algorithm for $d$ ranging from $10^4$ to $10^7$, always setting $k = \sqrt{d}$. The results are shown in Table 1. The mean estimate is always very close to the true value of $\varepsilon$. There is more noise when $d$ is small, but our method is primarily designed for "normal" scale contemporary deep-learning models which easily surpass 1M parameters, and at these sizes the method is very accurate.

| $\sigma$ | analytical $\varepsilon$ | $d = 10^4$ | $d = 10^5$ | $d = 10^6$ | $d = 10^7$ |
|---|---|---|---|---|---|
| 0.541 | 10.0 | $9.89 \pm 0.71$ | $10.1 \pm 0.41$ | $10.0 \pm 0.23$ | $10.0 \pm 0.10$ |
| 1.54 | 3.0 | $3.00 \pm 0.46$ | $3.00 \pm 0.31$ | $2.96 \pm 0.15$ | $3.00 \pm 0.08$ |
| 4.22 | 1.0 | $0.98 \pm 0.41$ | $1.05 \pm 0.23$ | $0.99 \pm 0.14$ | $1.00 \pm 0.07$ |

Table 1: One-shot auditing of the Gaussian mechanism for a range of values of $d$, setting $k = \sqrt{d}$ and $\delta = 10^{-6}$. For each value of $\varepsilon$, we set $\sigma$ using the optimal calibration of Balle and Wang (2018). Shown is the mean and std $\varepsilon_{\text{est}}$ over 50 simulations.

## 4 ONE-SHOT PRIVACY ESTIMATION FOR FL WITH RANDOM CANARIES

We now extend this idea to DP Federated Averaging to estimate the privacy of releasing the final model parameters in one shot, during model training. We propose adding $k$ canary clients to the training population who participate exactly as real clients do. Each canary client generates a random model update sampled from the unit sphere, which it returns at every round in which it participates, scaled to have norm equal to the clipping norm for the round. After training, we collect the set of canary/final-model cosines, fit them to a Gaussian, and compare them to the null hypothesis distribution $\mathcal{N}(0, 1/d)$ just as we did for the basic Gaussian mechanism. The procedure is described in Algorithm 2.

The rationale behind this attack is as follows. FL is an optimization procedure in which each client contributes an update at each round in which it participates, and each model iterate is a linear combination of all updates received thus far, plus Gaussian noise. Our threat model allows the

---

[2]To our knowledge, there is no way of confidently inferring that a set of samples comes from a given distribution, or even that they come from a distribution that is close to the given distribution in some metric. However it gives us some confidence to see that a strong goodness-of-fit test cannot rule out the given distribution even with a very high number of samples. The low significance level means the test is so sensitive it would be expected to reject even a set of truly Gaussian-distributed set of samples 15% of the time. This is a more quantitative claim than visually comparing a histogram or empirical CDF, as is commonly done.

---

**Algorithm 2** Privacy estimation via random canaries

---

1: **Input:** Client selection function **clients**, client training functions $\tau_j$, canary selection function **canaries**, set of canary updates $c_i$, number of rounds $T$, initial parameters $\theta_0$, noise generator $Z$, $\ell_2$ clip norm function $S$, privacy parameter $\delta$, server learning rate $\eta$
2: **for** $t = 1, \ldots, T$ **do**
3: $\quad \rho = \vec{0}$
4: $\quad$ **for** $j \in$ **clients**$(t)$ **do**
5: $\quad\quad \rho \leftarrow \rho + \text{CLIP}(\tau_j(\theta_{t-1}); S(t))$
6: $\quad$ **for** $i \in$ **canaries**$(t)$ **do**
7: $\quad\quad \rho \leftarrow \rho + \text{PROJ}(c_i; S(t))$
8: $\quad n = |\mathbf{clients}(t)| + |\mathbf{canaries}(t)|$
9: $\quad \theta_t \leftarrow \theta_{t-1} + \eta(\rho + Z(t))/n$
10: **for all canaries** $i$ **do**
11: $\quad g_i \leftarrow \langle c_i, \theta_T \rangle / \|\theta_T\|$
12: $\mu, \sigma \leftarrow \mathbf{mean}(\{g_i\}), \mathbf{std}(\{g_i\})$
13: $\varepsilon \leftarrow \varepsilon(\mathcal{N}(0, 1/d) \,\|\, \mathcal{N}(\mu, \sigma^2); \delta)$

14: **function** CLIP$(x; \kappa)$
15: $\quad$ **return** $x \cdot \min(1, \kappa/\|x\|)$

16: **function** PROJ$(x; \kappa)$
17: $\quad$ **return** $x \cdot \kappa/\|x\|$

---

attacker to control the updates of a client when it participates, and the ability to inspect the final model. We would argue that a powerful (perhaps optimal, under some assumptions) strategy is to return a very large update that is essentially orthogonal to all other updates, and then measure the dot product (or cosine) to the final model. Fortunately, the attacker does not even need any information about the other clients' updates in order to find such an orthogonal update: if the dimensionality of the model is high relative to the number of clients, randomly sampled canary updates are nearly orthogonal to all the true client updates and also to each other.

Unlike many works that only produce correct estimates when clients are sampled uniformly and independently at each round, our method makes no assumptions on the pattern of client participation. The argument of the preceding paragraph holds whether clients are sampled uniformly at each round, shuffled and processed in batches, or even if they are sampled according to the difficult-to-characterize *de facto* pattern of participation of real users in a production system. Our only assumption is that canaries can be inserted according to the same distribution that real clients are. In production settings, a simple and effective strategy would be to designate a small fraction of real clients to have their model updates replaced with the canary update whenever they participate. If the participation pattern is such that memorization is easier, for whatever reason, the distribution of canary/final-model cosines will have a higher mean, leading to higher $\varepsilon$ estimates.

The task of the adversary is to distinguish between canaries that were inserted during training vs. canaries that were not observed during training, based on observation of the cosine of the angle between the canary and the final model. If the canary was not inserted during training, we know by the argument in the preceding section that the distribution of the cosine will follow $\mathcal{N}(0, 1/d)$. To model the distribution of observed canary cosines, we approximate the distribution with a Gaussian with the same empirical mean $\hat{\mu}$ and variance $\hat{\sigma}^2$. Then we report the $\varepsilon$ computed by comparing two Gaussian distributions as described in Appendix A.

We stress that our empirical $\varepsilon$ estimate should not be construed as a formal bound on the worst-case privacy leakage. Rather, a low value of $\varepsilon$ can be taken as evidence that an adversary implementing this particular, powerful attack will have a hard time inferring the presence of any given user upon observing the final model. If we suppose that the attack is strong, or even optimal, then we can infer that *any* attacker will not be able to perform MI successfully, and therefore our $\varepsilon$ is a justifiable metric of the true privacy when the final model is released. Investigating conditions under which this could be proven would be a valuable direction for future work.

Existing analytical bounds on $\varepsilon$ for DP-SGD assume that all intermediate model updates can be observed by the adversary (Abadi et al., 2016). In cross-device FL, an attacker who controls multiple participating devices could in principle obtain some or all model checkpoints. But we argue there are at least two important cases where the "final-model-only" threat model is realistic. First, one can run DP-FedAvg on centrally collected data in the datacenter to provide a user-level DP guarantee. In this scenario, clients still entrust the server with their data, but intermediate states are ephemeral and only the final privatized model (whether via release of model parameters or black-box access) is made public. Second, there is much recent interest in using Trusted Execution Environments (TEEs)

---

**Algorithm 3** Privacy estimation via random canaries using all iterates

---

1: **Input:** As in Algorithm 2, but with *unobserved* canary updates $c_i^0$ and *observed* canary updates $c_i^1$.
2: **for** $t = 1, \ldots, T$ **do**
3:      $\rho = \vec{0}$
4:      **for** $j \in \mathbf{clients}(t)$ **do**
5:          $\rho \leftarrow \rho + \mathrm{CLIP}(\tau_j(\theta_{t-1}); S(t))$
6:      **for** $i \in \mathbf{canaries}(t)$ **do**
7:          $\rho \leftarrow \rho + \mathrm{PROJ}(c_i^1; S(t))$
8:      $n = |\mathbf{clients}(t)| + |\mathbf{canaries}(t)|$
9:      $\bar{\rho} \leftarrow (\rho + Z(t))/n$
10:      **for all canaries** $i$ **do**
11:          $g_{t,i}^0 = \langle c_i^0, \bar{\rho} \rangle / \|\bar{\rho}\|$
12:          $g_{t,i}^1 = \langle c_i^1, \bar{\rho} \rangle / \|\bar{\rho}\|$
13:      $\theta_t \leftarrow \theta_{t-1} + \eta \bar{\rho}$
14: **for all canaries** $i$ **do**
15:      $g_i^0 \leftarrow \max_t g_{t,i}^0$
16:      $g_i^1 \leftarrow \max_t g_{t,i}^1$
17: $\mu_0, \sigma_0 \leftarrow \mathbf{mean}(\{g_i^0\}), \mathbf{std}(\{g_i^0\})$
18: $\mu_1, \sigma_1 \leftarrow \mathbf{mean}(\{g_i^1\}), \mathbf{std}(\{g_i^1\})$
19: $\varepsilon \leftarrow \varepsilon(\mathcal{N}(\mu_0, \sigma_0^2) \,\|\, \mathcal{N}(\mu_1, \sigma_1^2); \delta)$

---

for further data minimization. For example, using TEEs on server and client, a client could prove to the server that they are performing local training as intended without being able to access the model parameters (Mo et al., 2021). Therefore we believe the final-model-only threat model is realistic and important, and will be of increasing interest in coming years as TEEs become more widely used.

Aside from quantifying the privacy of releasing only the final model, our method allows us to explore how privacy properties are affected by varying aspects of training for which we have no tight formal analysis. As an important example (which we explore in experiments) we consider how the estimate changes if clients are constrained to participate a fixed number of times.

We also propose a simple extension to our method that allows us to estimate $\varepsilon$ under the threat model where all model updates are observed. We use as the test statistic the *maximum over rounds* of the angle between the canary and the model delta at that round. A sudden increase of the angle cosine at a particular round is good evidence that the canary was present in that round. Unfortunately in this case we can no longer express in closed form the distribution of max-over-rounds cosine of an canary that did not participate in training, because it depends on the trajectory of partially trained models, which is task and model specific. Our solution is to sample a set of unobserved canaries that are never included in model updates, but we still keep track of their cosines with each model delta and finally take the max. We approximate both the distributions of observed and unobserved maximum canary/model-delta cosines using Gaussian distributions and compute the optimal $\varepsilon$. The pseudocode for this modified procedure is provided in Algorithm 3. We will see that this method provides estimates of $\varepsilon$ close to the analytical bounds under moderate amounts of noise, providing evidence that our attack is strong.

## 5   EXPERIMENTS

In this section we present the results of experiments estimating the privacy leakage while training a model on a large-scale public federated learning dataset: the stackoverflow word prediction data/model of Reddi et al. (2020).[3] The model is a word-based LSTM with 4.1M parameters. We train the model for 2048 rounds with 167 clients per round, where each of the $m$=341k clients participates in exactly one round, amounting to a single epoch over the data. We use the adaptive clipping method of Andrew et al. (2021). With preliminary manual tuning, we selected a client learning rate of $1.0$, server learning rate of $0.56$, and momentum of $0.9$ on the server for all experiments because this choice gives good performance over a range of levels of DP noise. We use 1k canaries for each set of cosines; experiments with intermediate iterates use 1k observed and 1k unobserved canaries. We fix $\delta = m^{-1.1}$. We consider noise multipliers[4] in the range 0.0496 to 0.2317, corresponding to analytical

---

[3]We present experimental results on the image dataset EMNIST in Appendix F. Code to reproduce experiments is available at `https://github.com/google-research/federated/tree/master/one_shot_epe`.

[4]The noise multiplier is the ratio of the noise to the clip norm. When adaptive clipping is used, the clip norm varies across rounds, and the noise scales proportionally.

| Noise | analytical $\varepsilon$ | $\varepsilon_{\text{lo}}$-all | $\varepsilon_{\text{est}}$-all | $\varepsilon_{\text{lo}}$-final | $\varepsilon_{\text{est}}$-final |
|---|---|---|---|---|---|
| 0 | $\infty$ | 6.240 | 45800 | 2.88 | 4.60 |
| 0.0496 | 300 | 6.238 | 382 | 1.11 | 1.97 |
| 0.0986 | 100 | 5.05 | 89.4 | 0.688 | 1.18 |
| 0.2317 | 30 | 0.407 | 2.693 | 0.311 | 0.569 |

Table 2: Comparing $\varepsilon$ estimates using all model deltas vs. using the final model only. $\varepsilon_{\text{lo}}$ is the empirical 95% lower bound from our modified Jagielski et al. (2020) method. For moderate noise, $\varepsilon_{\text{est}}$-all is in the ballpark of the analytical $\varepsilon$, providing evidence that the attack is strong and therefore the $\varepsilon$ estimates are reliable. On the other hand, $\varepsilon_{\text{est}}$-final is far lower, indicating that when the final model is observed, privacy is better.

$\varepsilon$ estimates from 300 down to 30.[5] We also include experiments with clipping only (noise multiplier is 0). Table 3 in Appendix D shows that across the range of noise multipliers, the participation of 1k canaries had no significant impact on model accuracy – at most causing a 0.1% absolute decrease.

We also report a high-probability lower bound on $\varepsilon$ that comes from applying a modified version of the method of Jagielski et al. (2020) to the set of cosines. That work uses Clopper-Pearson upper bounds on the achievable FPR and FNR of a thresholding classifier to derive a bound on $\varepsilon$. We make two changes: following Zanella-Béguelin et al. (2023), we use the tighter and more centered Jeffreys confidence interval for the upper bound on FNR at some threshold $a$, and we use the exact CDF of the null distribution for the FPR as described in Section 4. We refer to this lower bound as $\varepsilon_{\text{lo}}$. We set $\alpha = 0.05$ to get a 95%-confidence bound. This estimate has a maximum posisble value of 6.24 with perfect separation. We include it to illustrate that our method does not suffer from this limitation, and to show that our estimate is not falsified by the lower bound.

We first consider the case where the intermediate updates are released as described in Algorithm 3. The middle columns of Table 2 shows the results of these experiments over a range of noise multipliers. For the lower noise multipliers, our method easily separates the cosines of observed vs. unobserved canaries, producing very high estimates $\varepsilon_{\text{est}}$-all, which are much higher than lower bounds $\varepsilon_{\text{lo}}$-all estimated by previous work. This confirms our intuition that intermediate model updates give the adversary significant power to detect the presence of individuals in the data. It also provides evidence that the canary cosine attack is strong, increasing our confidence that the $\varepsilon$ estimates assuming a weakened adversary that observes only the final model is not a severe underestimate.

The rightmost columns of Table 2 show the results of restricting the adversary to observe only the final model, as described in Algorithm 2. Now $\varepsilon_{\text{est}}$ is significantly smaller than when the adversary has access to all intermediary updates. With clipping only, our estimate is 4.60, which is still quite weak from a rigorous privacy perspective.[6] But with even a small amount of noise, we approach the high-privacy regime of $\varepsilon \sim 1$, confirming observations of practitioners that a small amount of noise is sufficient to prevent most memorization.

## 5.1 Experiments with multiple canary presentations

Here we highlight the ability of our method to estimate privacy when we vary not only the threat model, but also aspects of the training algorithm that may reasonably be expected to change privacy properties, but for which no tight analysis has been obtained. We consider presenting each canary a fixed multiple number of times, modeling the scenario in which clients are only allowed to check in for training every so often. In practice, a client need not participate in every period, but to obtain worst-case estimates, we present the canary in every period.

In Figure 1 we show kernel density estimation plots of the canary cosine sets. As the number of presentations increases in each plot, the distributions become more and more clearly separated.

---

[5]This bound comes from assuming the adversary knows on which round a target user participated. It is equal to the privacy loss of the unamplified Gaussian mechanism applied once with no composition. Stronger guarantees might come from assuming shuffling of the clients (Feldman et al., 2023; 2022), but tight bounds for that case are not known.

[6]An $\varepsilon$ of 5 means that an attacker can go from a small suspicion that a user participated (say, 10%) to a very high degree of certainty (94%) (Desfontaines, 2018).

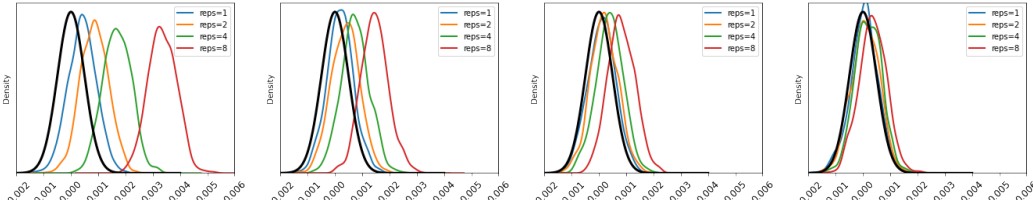

Figure 1: Density plots of cosine values with four values of noise corresponding to analytical epsilons ($\infty$, 300, 100, 30) and four values of canary repetitions (1, 2, 4, 8). The black curve in each plot is the pdf of the null distribution $\mathcal{N}(0, 1/d)$. With no noise ($\varepsilon = \infty$), the distributions are easily separable, with increasing separation for more canary repetitions. At higher levels of noise, distributions are less separable, even with several repetitions.

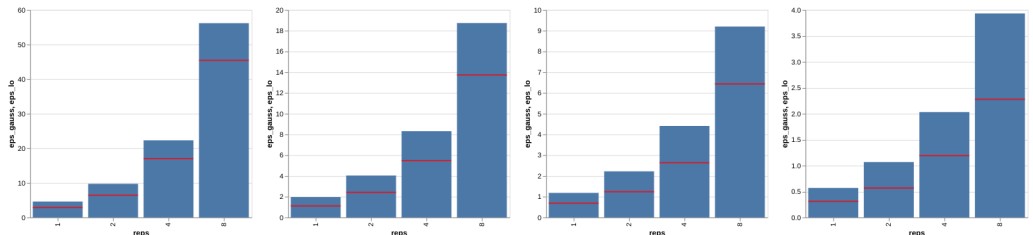

Figure 2: Blue bars are our $\varepsilon_{\text{est}}$ and red ticks are the $\varepsilon_{\text{lo}}$ 95%-confidence lower bound for four values of noise corresponding to analytical epsilons ($\infty$, 300, 100, 30) and four values of canary repetitions (1, 2, 4, 8). Note the difference of y-axis scales in each plot. Our estimate of epsilon increases sharply with the number of canary repetitions, confirming that limiting client participation improves privacy.

On the other hand as the amount of noise increases across the three plots, they converge to the null distribution. Also visible on this figure is that the distributions are roughly Gaussian-shaped, justifying the Gaussian approximation that is used in our estimation method. In Appendix E we give quantitative evidence for this observation. Finally we compare $\varepsilon_{\text{lo}}$ to our $\varepsilon_{\text{est}}$ with multiple canary presentations in Figure 2. For each noise level, $\varepsilon_{\text{est}}$ increases dramatically with increasing presentations, confirming our intuition that seeing examples multiple times dramatically reduces privacy. In Appendix F we provide analogous experiments on the federated EMNIST dataset, with similar results.

## 6 CONCLUSION

In this work we have introduced a novel method for empirically estimating the privacy loss during training of a model with DP-FedAvg. For natural production-sized problems (millions of parameters, hundreds of thousands of clients), it produces reasonable privacy estimates during the same single training run used to estimate model parameters, without significantly degrading the utility of the model, and does not require any prior knowledge of the task, data or model. The resulting $\varepsilon_{\text{est}}$ can be interpreted as bounding the degree of confidence that a particular strong adversary could have in performing membership inference. It gives a reasonable metric for comparing how privacy loss changes between arbitrary variants of client-participation, or other variations of DP-FedAvg for which no method for producing a tight analytical estimate of $\varepsilon$ is known.

In future work we would like to explore how our metric is related to formal bounds on the privacy loss. We would also like to open the door to empirical refutation of our epsilon metric – that is, for researchers to attempt to design a successful attack on a training mechanism for which our metric nevertheless estimates a low value of epsilon. To the extent that we are confident no such attack exists, we can be assured that our estimate is faithful. We note that this is the case with techniques like the cryptographic hash function SHA-3: although no proof exists that inverting SHA-3 is computationally difficult, it is used in high-security applications because highly motivated security experts have not been able to mount a successful inversion attack in the many years of its existence.

## 7  ACKNOWLEDGEMENTS

We are extremely grateful to Krishna Pillutla for his detailed review of an early draft of the paper. We also thank the anonymous ICLR reviewers who helped us to make our proofs more rigorous.

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

# A  ALGORITHM FOR EXACT COMPUTATION OF $\varepsilon$ COMPARING TWO GAUSSIAN DISTRIBUTIONS

In this section we give the details of the computation for estimating $\varepsilon$ when $A(D)$ and $A(D')$ are both Gaussian-distributed with different variances. An implementation of this computation is available at `https://github.com/google-research/federated/tree/master/one_shot_epe/empirical_privacy_estimation_lib.py`.

Let the distribution under $A(D)$ be $P_1 = \mathcal{N}(\mu_1, \sigma_1^2)$ and the distribution under $A(D')$ be $P_2 = \mathcal{N}(\mu_2, \sigma_2^2)$ with densities $p_1$ and $p_2$ respectively. Define $f_{P_1||P_2}(x) = \log \frac{p_1(x)}{p_2(x)}$. Now $Z_1 = f_{P_1||P_2}(X_1)$ with $X_1 \sim P_1$ is the privacy loss random variable. Symmetrically define $Z_2 = f_{P_2||P_1}(X_2)$ with $X_2 \sim P_2$.

From Steinke (2022) Prop. 7 we have that $(\epsilon, \delta)$-DP implies

$$\Pr[Z_1 > \varepsilon] - e^\varepsilon \Pr[-Z_2 > \varepsilon] \le \delta \text{ and } \Pr[Z_2 > \varepsilon] - e^\varepsilon \Pr[-Z_1 > \varepsilon] \le \delta.$$

Now we can compute

$$\begin{aligned}
f_{P_1||P_2}(x) &= \log \frac{p_1(x)}{p_2(x)} \\
&= \log\left(\frac{\sigma_2}{\sigma_1} \exp\left(-\frac{1}{2}\left[\frac{(x-\mu_1)^2}{\sigma_1^2} - \frac{(x-\mu_2)^2}{\sigma_2^2}\right]\right)\right) \\
&= \log \sigma_2 - \log \sigma_1 + \frac{(x-\mu_2)^2}{2\sigma_2^2} - \frac{(x-\mu_1)^2}{2\sigma_1^2} \\
&= ax^2 + bx + c
\end{aligned}$$

where

$$\begin{aligned}
a &= \frac{1}{2}\left(\frac{1}{\sigma_2^2} - \frac{1}{\sigma_1^2}\right), \\
b &= \frac{\mu_1}{\sigma_1^2} - \frac{\mu_2}{\sigma_2^2}, \\
\text{and } c &= \frac{1}{2}\left(\left(\frac{\mu_2}{\sigma_2}\right)^2 - \left(\frac{\mu_1}{\sigma_1}\right)^2\right) + \log \sigma_2 - \log \sigma_1.
\end{aligned}$$

To compute $\Pr[Z_1 > \varepsilon]$, we need $\Pr[aX_1^2 + bX_1 + (c - \varepsilon) > 0]$ with $X_1 \sim P_1$. To do so, divide the range of $X_1$ into intervals according to the zeros of $R(x) = ax^2 + bx + (c - \varepsilon)$. For example, if $R$ has roots $r_1 < r_2$ and $a$ is positive, we can compute $\Pr[Z_1 > \varepsilon] = \Pr[X_1 < r_1] + \Pr[X_1 > r_2]$, using the CDF of the Normal distribution. This requires considering a few cases, depending on the sign of $a$ and the sign of the determinant $b^2 - 4a(c - \varepsilon)$. Now note that $f_{P_2||P_1} = -f_{P_1||P_2}$, so

$$\Pr[-Z_2 > \varepsilon] = \Pr[-f_{P_2||P_1}(X_2) > \varepsilon] = \Pr[aX_2^2 + bX_2 + (c - \varepsilon) > 0].$$

So the two events we are interested in ($Z_1 > \varepsilon$ and $-Z_2 > \varepsilon$) are the same, only when we compute their probabilities according to $P_1$ vs. $P_2$ we use different values for $\mu$ and $\sigma$.

For numerical stability, the probabilities should be computed in the log domain. So we get

$$\begin{aligned}
\log \delta &\ge \log\left(\Pr[Z_1 > \varepsilon] - e^\varepsilon \Pr[-Z_2 > \varepsilon]\right) \\
&= \log \Pr[Z_1 > \varepsilon] + \log\left(1 - \exp(\varepsilon + \log \Pr[-Z_2 > \varepsilon] - \log \Pr[Z_1 > \varepsilon])\right).
\end{aligned}$$

Note it can happen that $\Pr[Z_1 > \varepsilon] < e^\varepsilon \Pr[-Z_2 > \varepsilon]$ in which case the corresponding bound is invalid. A final trick we suggest for numerical stability is if $X \sim \mathcal{N}(\mu, \sigma^2)$ to use $\Pr(X < \mu; t, \sigma^2)$ in place of $\Pr(X > t; \mu, \sigma^2)$.

Now to determine $\varepsilon$ at a given target $\delta$, one can perform a line search over $\varepsilon$ to find the value that matches.

## B  PROOFS OF RESULTS FROM THE MAIN TEXT

**Preliminaries.**  First we must introduce an elementary result about the measure of a hyperspherical cap. In $\mathbb{R}^d$, the area ($(d-1)$-measure) of the hypersphere of radius $r$ is (Li, 2011)

$$A_d(r) = \frac{2\pi^{d/2}}{\Gamma(d/2)} r^{d-1}.$$

Define the unit hyperspherical cap of maximal angle $\theta \in [0, \pi]$ to be the set $\{x : \|x\| = 1 \text{ and } x_1 \geq \cos\theta\}$, and let $M_d(\theta)$ denote its area. From (Li, 2011) we have

$$M_d(\theta) = \int_0^\theta A_{d-1}(\sin\phi) \, d\phi,$$

from which it follows from the fundamental theorem of calculus that

$$\frac{d}{d\theta} M_d(\theta) = A_{d-1}(\sin\theta) = \frac{2\pi^{\frac{d-1}{2}}}{\Gamma(\frac{d-1}{2})} \sin^{d-2}\theta. \tag{2}$$

**Proposition 3.1.** *For $d \in \mathbb{N}, d \geq 2$, let $c$ be sampled uniformly from $\mathbb{S}^{d-1}$, and let $\tau_d = \langle c, v\rangle / \|v\| \in [-1, 1]$ be the cosine similarity between $c$ and some arbitrary independent nonzero vector $v$. Then, the probability density function of $\tau_d$ is*

$$f_d(t) = \frac{\Gamma(\frac{d}{2})}{\Gamma(\frac{d-1}{2})\sqrt{\pi}} (1 - t^2)^{\frac{d-3}{2}}.$$

*Proof.* Due to spherical symmetry, without loss of generality, we can take $v$ to be constant and equal to the first standard basis vector $e_1$. First we describe the distribution of the angle $\theta \in [0, \pi]$ between $c$ and $e_1$, then change variables to get the distribution of its cosine $\tau_d$. Using (2) and normalizing by the total area of the sphere $A_d(1)$, the density of the angle is

$$\phi_d(\theta) = (A_d(1))^{-1} \frac{d}{d\theta} M_d(\theta)$$

$$= \left(\frac{2\pi^{\frac{d}{2}}}{\Gamma(\frac{d}{2})}\right)^{-1} \frac{2\pi^{\frac{d-1}{2}}}{\Gamma(\frac{d-1}{2})} \sin^{d-2}\theta$$

$$= \frac{\Gamma(\frac{d}{2})}{\Gamma(\frac{d-1}{2})\sqrt{\pi}} \sin^{d-2}\theta.$$

Now change variables to find the density of the angle cosine $\tau_d = \cos(\theta) \in [-1, 1]$:

$$f_d(t) = \phi_d(\arccos t) \cdot \left| \frac{d}{dt} \arccos(t) \right|$$

$$= \frac{\Gamma(\frac{d}{2})}{\Gamma(\frac{d-1}{2})\sqrt{\pi}} [\sin(\arccos t)]^{d-2} \left| -\frac{1}{\sqrt{1-t^2}} \right|$$

$$= \frac{\Gamma(\frac{d}{2})}{\Gamma(\frac{d-1}{2})\sqrt{\pi}} \frac{\left(\sqrt{1-t^2}\right)^{d-2}}{\sqrt{1-t^2}}$$

$$= \frac{\Gamma(\frac{d}{2})}{\Gamma(\frac{d-1}{2})\sqrt{\pi}} (1 - t^2)^{\frac{d-3}{2}}.$$

$\square$

**Proposition 3.2.** *Let $\tau_d$ be the random variable of Proposition 3.1 in $\mathbb{R}^d$. Then $\tau_d\sqrt{d}$ converges in distribution to $\mathcal{N}(0, 1)$ as $d \to \infty$, i.e., $\forall \lambda \in \mathbb{R}$, $\lim_{d\to\infty} \mathbb{P}(\tau_d \leq \lambda/\sqrt{d}) = \mathbb{P}_{Z \sim \mathcal{N}(0,1)}(Z \leq \lambda)$.*

*Proof.* The probability density function of $\tau_d\sqrt{d}$ is

$$\hat{f}_d(t) = \begin{cases} \frac{\Gamma(\frac{d}{2})}{\Gamma(\frac{d-1}{2})\sqrt{\pi d}} \left(1 - t^2/d\right)^{\frac{d-3}{2}} & \text{for } |t| \leq \sqrt{d}; \\ 0 & \text{for } |t| \geq \sqrt{d}. \end{cases}$$

For any $t$, for large enough $d$, $t \leq \sqrt{d}$, so

$$\lim_{d\to\infty} \hat{f}_d(t) = \left( \lim_{d\to\infty} \frac{\Gamma(\frac{d}{2})}{\Gamma(\frac{d-1}{2})\sqrt{\pi d}} \right) \cdot \left( \lim_{d\to\infty} \left(1 - t^2/d\right)^{\frac{d}{2}} \right) \cdot \left( \lim_{d\to\infty} \left(1 - t^2/d\right)^{-\frac{3}{2}} \right)$$

$$= \frac{1}{\sqrt{2\pi}} \cdot e^{-t^2/2} \cdot 1,$$

where we have used the fact that $\frac{\Gamma(\frac{d}{2})}{\Gamma(\frac{d-1}{2})} \sim \sqrt{d/2}$. The result follows by Scheffe's theorem, which states that pointwise convergence of the density function implies convergence in distribution (Scheffé, 1947). □

**Lemma B.1.** *If $\tau_d$ is distributed according to the cosine angle distribution described in Proposition 3.1, then $Var[\tau_d] = 1/d$.*

*Proof.* Let $c = (c_1, \dots, c_d)$ be uniform on $\mathbb{S}^{d-1}$. Then $c_1 = \langle c, e_1 \rangle$ has the required distribution. $\mathbb{E}[c_1]$ is zero, so we are interested in $Var[c_1] = \mathbb{E}[c_1^2]$. Since $\sum_i c_i^2 = 1$, we have that $\mathbb{E}[\sum_i c_i^2] = \sum_i \mathbb{E}[c_i^2] = 1$. But all of the $c_i$ have the same distribution, so $\mathbb{E}[c_1^2] = 1/d$. □

**Lemma B.2.** *If $c$ is sampled uniformly from $\mathbb{S}^{d-1}$, then for any $x, y \in \mathbb{R}^d$, we have that $\mathbb{E}[\langle c, x \rangle \langle c, y \rangle] = \langle x, y \rangle / d$.*

*Proof.* Let $c = (c_1, \dots, c_d)$. From the proof of Lemma B.1 we have that for all $i$, $\mathbb{E}[c_i^2] = 1/d$. It is also easy to see that for all $i \neq j$, $\mathbb{E}[c_i c_j] = 0$, since by symmetry $\mathbb{E}[c_i c_j] = \mathbb{E}[(-c_i)c_j] = -\mathbb{E}[c_i c_j]$. Therefore,

$$\mathbb{E}[\langle c, x \rangle \langle c, y \rangle] = \sum_{i,j} x_i y_j \mathbb{E}[c_i c_j] = \sum_i \frac{x_i y_i}{d} = \frac{\langle x, y \rangle}{d}.$$

□

**Theorem 3.3.** *For $d \in \mathbb{N}$. For some $m$ (which may grow with $d$), let $x_{d1} \dots x_{dm}$ such that $\|X_d\| = o(\sqrt{d})$ where $X_d = \sum_{j=1}^m x_{dj}$. Let $k = o(d)$, but $k = \omega(1)$, and for $i \in [k]$, let $c_{di}$ be sampled i.i.d. uniformly from $\mathbb{S}^{d-1}$. Let $Z_d \sim \mathcal{N}(0; I_d)$. For $\sigma \in \mathbb{R}^+$, define the mechanism result $\rho_d = X_d + \sum_{i=1}^k c_{di} + \sigma Z_d$, and the cosine values $g_{di} = \frac{\langle c_{di}, \rho_d \rangle}{\|\rho_d\|}$. Write the empirical mean of the cosines $\hat{\mu}_d = \frac{1}{k} \sum_{i=1}^k g_{di}$, and the empirical variance $\hat{\sigma}_d^2 = \frac{1}{k} \sum_{i=1}^k (g_{di} - \hat{\mu}_d)^2$. Let $\hat{\varepsilon}_d = \varepsilon(\mathcal{N}(0, 1/d) \| \mathcal{N}(\hat{\mu}_d, \hat{\sigma}_d^2); \delta)$. Then $\hat{\varepsilon}_d$ converges in probability to the constant $\varepsilon(\mathcal{N}(0, \sigma^2) \| \mathcal{N}(1, \sigma^2); \delta)$.*

*Proof.* We will show that as $d \to \infty$, we have that $\sqrt{d}\hat{\mu}_d \overset{p}{\longrightarrow} 1/\sigma$ and $d\hat{\sigma}_d^2 \overset{p}{\longrightarrow} 1$. Note that $\hat{\varepsilon}_d = \varepsilon(\mathcal{N}(0, 1) \| \mathcal{N}(\sqrt{d}\hat{\mu}_d, d\hat{\sigma}_d^2); \delta)$, since this is just a rescaling of both distributions by a factor of $1/\sqrt{d}$, which does not change $\varepsilon$. Therefore by the Mann-Wald theorem (Mann and Wald, 1943), the estimate $\hat{\varepsilon}_d$ converges in probability to $\varepsilon(\mathcal{N}(0, 1) \| \mathcal{N}(1/\sigma, 1); \delta)$. Now these two distributions are just a scaling of $A(D) \sim \mathcal{N}(0, \sigma^2)$ and $A(D') \sim \mathcal{N}(1, \sigma^2)$ by a factor of $1/\sigma$. This proves our claim.

For the remainder of the proof, we will omit the subscript $d$ on $X_d$, $c_{di}$, $Z_d$, and $\rho_d$. All summations, such as $\sum_i c_i$, should be understood as going from 1 to $k$.

Rewrite

$$\sqrt{d}\hat{\mu}_d = \sqrt{d}\left(\frac{1}{k} \sum_i \frac{\langle c_i, \rho \rangle}{\|\rho\|}\right) = \left(\frac{\|\rho\|}{\sqrt{d}}\right)^{-1} \left(\frac{1}{k} \sum_i \langle c_i, \rho \rangle\right).$$

We will show that $\frac{\|\rho\|^2}{d} \overset{p}{\longrightarrow} \sigma^2$, while $\frac{1}{k} \sum_i \langle c_i, \rho \rangle \overset{p}{\longrightarrow} 1$.

We will need to consider the following dot products: $\langle X, c_i \rangle$ (for all $i$), $\langle X, Z \rangle$, $\langle c_i, Z \rangle$ (for all $i$), $\langle c_i, c_j \rangle$ (for all $i < j$), and $\langle Z, Z \rangle = \|Z\|^2$. All but $\|Z\|^2$ are zero-mean, and they are pairwise uncorrelated.[7] It follows that $\langle c_i, \rho \rangle$ and $\langle c_j, \rho \rangle$ are uncorrelated for all $i \neq j$. Note that $\langle c_i, Z \rangle \sim \mathcal{N}(0, 1)$, and $\langle c_i, c_j \rangle$ is distributed like $\tau_d$ (and $\langle X, c_i \rangle$ is distributed like $\|X\|\tau_d$). Finally, note that $\langle c_i, Z \rangle^2$ is Chi-squared distributed with a single degree of freedom while $\langle Z, Z \rangle = \|Z\|^2$ is Chi-squared distributed with $d$ degrees of freedom.

---

[7]To see that $\langle X, Z \rangle$ and $\|Z\|^2$ are uncorrelated, write $\mathbb{E}[\langle X, Z \rangle \|Z\|^2] = \|X\| \cdot \mathbb{E}[\langle e_1, Z \rangle \|Z\|^2] = \|X\| \left( \mathbb{E}[Z_1^3] + \sum_{i=2}^d \mathbb{E}[Z_1 Z_i^2] \right) = 0 = \mathbb{E}[\langle X, Z \rangle]$, since the $Z_i$ are independent draws from $\mathcal{N}(0, 1)$.

Now we can proceed,

$$\mathbb{E}\left[\frac{||\rho||^2}{d}\right] = \frac{1}{d}\mathbb{E}\left[\left\langle X + \sum_i c_i + \sigma Z,\ X + \sum_i c_i + \sigma Z\right\rangle\right]$$

$$= \frac{1}{d}\left(\|X\|^2 + k + \sigma^2 d\right)$$

$$= \sigma^2 + o(1).$$

The variance decomposes:

$$\text{Var}\left[\frac{||\rho||^2}{d}\right] = \frac{1}{d^2}\text{Var}\left[\left\langle X + \sum_i c_i + \sigma Z,\ X + \sum_i c_i + \sigma Z\right\rangle\right]$$

$$= \frac{1}{d^2}\left(\begin{array}{c}\sum_i \text{Var}\left[2\langle X, c_i\rangle\right] + \text{Var}[2\langle X, \sigma Z\rangle] + \sum_{i<j}\text{Var}\left[2\langle c_i, c_j\rangle\right] \\ + \sum_i \text{Var}\left[2\langle c_i, \sigma Z\rangle\right] + \text{Var}[\langle \sigma Z, \sigma Z\rangle]\end{array}\right)$$

$$= \frac{1}{d^2}\left(4k\frac{\|X\|^2}{d} + 4\sigma^2\|X\|^2 + \frac{2k(k-1)}{d} + 4k\sigma^2 + 2\sigma^4 d\right),$$

$$= o(1).$$

Taken together, these imply that $\frac{||\rho||}{\sqrt{d}} \xrightarrow{p} \sigma$.

Now reusing many of the same calculations,

$$\mathbb{E}\left[\frac{1}{k}\sum_i\langle c_i, \rho\rangle\right] = \frac{1}{k}\sum_i\mathbb{E}\left[\langle c_i, X + \sum_j c_j + \sigma Z\rangle\right]$$

$$= \frac{1}{k}\left(\sum_i\mathbb{E}[\langle c_i, X\rangle] + \sum_{i,j}\mathbb{E}[\langle c_i, c_j\rangle] + \sigma\sum_i\mathbb{E}[\langle c_i, Z\rangle]\right)$$

$$= \frac{1}{k}\left(0 + k + 0\right) = 1,$$

and

$$\text{Var}\left[\frac{1}{k}\sum_i\langle c_i, \rho\rangle\right] = \frac{1}{k^2}\sum_i\left(\text{Var}\left[\langle c_i, X\rangle\right] + \sum_j\text{Var}\left[\langle c_i, c_j\rangle\right] + \sigma^2\text{Var}\left[\langle c_i, Z\rangle\right]\right)$$

$$= \frac{1}{k}\left(\frac{\|X\|^2}{d} + \frac{k-1}{d} + \sigma^2\right)$$

$$= o(1),$$

which together imply that $\frac{1}{k}\sum_i\langle c_i, \rho\rangle \xrightarrow{p} 1$.

Now consider

$$d\hat{\sigma}_d^2 = d\left(\frac{1}{k}\sum_i g_i^2 - \left(\frac{1}{k}\sum_i g_i\right)^2\right)$$

$$= \frac{d}{||\rho||^2}\left(\frac{1}{k}\sum_i\langle c_i, \rho\rangle^2 - \left(\frac{1}{k}\sum_i\langle c_i, \rho\rangle\right)^2\right).$$

We already have that $\frac{d}{||\rho||^2} \xrightarrow{p} \frac{1}{\sigma^2}$ and $\frac{1}{k}\sum_i\langle c_i, \rho\rangle \xrightarrow{p} 1$, so to demonstrate that $d\hat{\sigma}_d^2 \xrightarrow{p} 1$, it will be sufficient to show $\frac{1}{k}\sum_i\langle c_i, \rho\rangle^2 \xrightarrow{p} 1 + \sigma^2$. To that end:

$$\mathbb{E}\left[\frac{1}{k}\sum_i\langle c_i, \rho\rangle^2\right] = \frac{1}{k}\sum_i\mathbb{E}\left[\left(\langle c_i, X\rangle + \sum_j\langle c_i, c_j\rangle + \langle c_i, \sigma Z\rangle\right)^2\right]$$

$$= \frac{1}{k}\sum_i\left(\mathbb{E}[\langle c_i, X\rangle^2] + \sum_j\mathbb{E}[\langle c_i, c_j\rangle^2] + \mathbb{E}[\langle c_i, \sigma Z\rangle^2]\right)$$

$$= \frac{1}{k}\sum_i\left(\frac{\|X\|^2}{d} + \left(1 + \frac{k-1}{d}\right) + \sigma^2\right)$$

$$= 1 + \sigma^2 + o(1).$$

Finally, to show that $\mathrm{Var}\left(\frac{1}{k}\sum_i\langle c_i, \rho\rangle^2\right) = o(1)$, we will show that

$$\mathbb{E}\left[\left(\frac{1}{k}\sum_i\langle c_i, \rho\rangle^2\right)^2\right] = \mathbb{E}\left[\frac{1}{k}\sum_i\langle c_i, \rho\rangle^2\right]^2 + o(1).$$

Rewrite

$$\mathbb{E}\left[\left(\frac{1}{k}\sum_i\langle c_i, \rho\rangle^2\right)^2\right] = \frac{1}{k^2}\sum_{i,j}\mathbb{E}[\langle c_i, \rho\rangle^2\langle c_j, \rho\rangle^2]$$

$$= \frac{1}{k^2}\sum_{i,j}\sum_{A,B,C,D}\mathbb{E}[\langle c_i, A\rangle\langle c_i, B\rangle\langle c_j, C\rangle\langle c_j, D\rangle]$$

where $A, B, C, D$ range independently over the terms of $\rho$: $\{X, c_1, \ldots, c_k, \sigma Z\}$. Now if $\{c_i, c_j\} \cap \{A, B, C, D\} = \emptyset$, then defining $V_{-ij} = \{c_1, \ldots, c_k, Z\} \setminus \{c_i, c_j\}$ and using Lemma B.2:

$$\mathbb{E}[\langle c_i, A\rangle\langle c_i, B\rangle\langle c_j, C\rangle\langle c_j, D\rangle] = \mathbb{E}[\mathbb{E}[\langle c_i, A\rangle\langle c_i, B\rangle\langle c_j, C\rangle\langle c_j, D\rangle | V_{-ij}]]$$

$$= \mathbb{E}[\mathbb{E}[\langle c_i, A\rangle\langle c_i, B\rangle | V_{-ij}]\,\mathbb{E}[\langle c_j, C\rangle\langle c_j, D\rangle | V_{-ij}]]$$

$$= \mathbb{E}[\langle A, B\rangle\langle C, D\rangle]/d^2$$

$$= \mathbb{E}[\langle A, B\rangle]\,\mathbb{E}[\langle C, D\rangle]/d^2$$

$$= \mathbb{E}[\langle c_i, A\rangle\langle c_i, B\rangle]\,\mathbb{E}[\langle c_j, C\rangle\langle c_j, D\rangle].$$

If we allow $|\{c_i, c_j\} \cap \{A, B, C, D\}| > 0$, there is only one case where the decomposition does not hold: if $A = B = c_j$ and $C = D = c_i$ (and $i \neq j$) we have $\mathbb{E}[\langle c_i, c_j\rangle^4] \sim 3/d^2$ (considering the fourth central moment of the standard Gaussian that $\sqrt{d}\langle c_i, c_j\rangle$ converges to by Proposition 3.2) whereas $\mathbb{E}[\langle c_i, c_j\rangle^2]^2 = 1/d^2$. Now we can conclude:

$$\mathbb{E}\left[\left(\frac{1}{k}\sum_i\langle c_i, \rho\rangle^2\right)^2\right] \sim \frac{1}{k^2}\sum_{i,j}\sum_{A,B,C,D}\mathbb{E}[\langle c_i, A\rangle\langle c_i, B\rangle]\,\mathbb{E}[\langle c_j, C\rangle\langle c_j, D\rangle] + \frac{k-1}{k}\frac{2}{d^2}$$

$$= \left(\frac{1}{k}\sum_i\sum_{A,B}\mathbb{E}[\langle c_i, A\rangle\langle c_i, B\rangle]\right)^2 + o(1)$$

$$= \mathbb{E}\left[\frac{1}{k}\sum_i\langle c_i, \rho\rangle^2\right]^2 + o(1).$$

$\square$

**Note on the rate of growth of $X_d$ in Theorem 3.3.** Theorem 3.3 requires that $\|X_d\| = o(\sqrt{d})$. This follows from any of several natural sufficient conditions:

- the data $x_{dj}$ are bounded and $m = o(\sqrt{d})$,
- the data $x_{dj}$ are i.i.d. from some isotropic distribution with finite variance and $m = o(d)$,
- there exists some constant $C \in \mathbb{R}^+$ with $\|X_d\|_\infty \leq C$.

This last condition is reasonable if the $x_{dj}$ are updates during some well-behaved learning process.

| noise multiplier | analytical $\varepsilon$ | baseline accuracy | accuracy 1k canaries added |
|---|---|---|---|
| 0 | $\infty$ | 25.3% | 25.3% |
| 0.064 | 194 | 24.0% | 23.9% |
| 0.102 | 93.8 | 23.1% | 23.1% |
| 0.184 | 40.2 | 21.5% | 21.5% |
| 0.234 | 28.9 | 20.6% | 20.5% |

Table 3: Comparison of accuracy of word prediction models trained with and without the presence of 1000 random canary clients. Inserting 1000 random clients among the 341k real clients in the Stackoverflow word prediction task has an almost negligible effect on model performance.

## C  Asymptotic Gaussianity of cosine statistics in Algorithm 1

Algorithm 1 specifies that we should fit the cosine statistics $g_i$ to a Gaussian distribution according to their empirical mean and variance. Empirical results, including visual comparison of the histogram to a Gaussian distribution function, and Anderson-Darling tests as we ran in Appendix E for the statistics of Algorithm 2, indicate that the empirical distribution of the cosine statistics in Algorithm 1 are approximately Gaussian in high dimensions. However it is not immediately clear how to prove this. Note that it would not be sufficient to show that each cosine sample $g_i$ is marginally Gaussian, which would be relatively easy. If we proved only that much, it could for example be the case that all of the samples $g_i$ are always identical to each other on each run (but Gaussian distributed across runs) so that the empirical variance on any run is zero. Then our method would be broken. We would need a stronger statement about their joint distribution, such as that the joint distribution converges to an isotropic Gaussian. While we believe this to be the case, we emphasize that it is not essential that the cosine statistics $g_i$ actually *be* independently Gaussian distributed in order for Algorithm 1 to be correct. Theorem 3.3 proves that we asymptotically recover the correct $\varepsilon$ by fitting them to a Gaussian, regardless of their true distribution.

## D  Impact of canaries on accuracy

One might worry that the adding random canary clients could impact model utility. In our experiments on Stackoverflow, with 341k real clients, the addition of 1000 canary clients had a negligible impact on model accuracy. The change in performance is shown in Table 3. We note that many practical FL problems have even more clients than this: see for example Xu et al. (2023) which trains language models using DP-FL in a variety of languages, almost of all of which have more than 3M clients each – ten times more than in our experiments.

## E  Gaussianity of cosine statistics from experiments

The density plots of the cosine statistics from the experiments in section 5.1, shown in Figure 1, appear Gaussian-shaped. To our knowledge, there is no way of confidently inferring that a set of samples comes from a given distribution, or even that they come from a distribution that is close to the given distribution in some metric. To quantify the error of our approximation of the cosine statistics with a Gaussian distribution, we apply the Anderson-Darling test to each set of cosines in Table 4 (Anderson and Darling, 1952). It gives us some confidence to see that a strong goodness-of-fit test cannot rule out that the distributions are Gaussian. This is a more quantitative claim than visually comparing the density.

## F  Supplementary experimental results on EMNIST dataset

In the main paper we presented results on the Stackoverflow federated word prediction task. Here we present similar results on the EMNIST character recognition dataset. It contains 814k characters written by 3383 users. The model is a CNN with 1.2M parameters. The users are shuffled and we train for five epochs with 34 clients per round. The optimizers on client and server are both SGD,

| Noise | 1 rep | 2 reps | 3 reps | 5 reps | 10 reps |
|---|---|---|---|---|---|
| 0 | 0.464 | **0.590** | 0.396 | 0.407 | 0.196 |
| 0.1023 | **0.976** | 0.422 | 0.340 | 0.432 | 0.116 |
| 0.2344 | 0.326 | 0.157 | 0.347 | **0.951** | 0.401 |

Table 4: Anderson-Darling test statistics for each set of canary-cosine samples from the experiments in section 5.1. The test rejects at a 1% significance level if the statistic is greater than 1.088, and rejects at 15% significance if the statistic is greater than 0.574. If all 15 (independent) distributions were Gaussian, the probability of observing three or more values with test statistic greater than 0.574 would be 68%, so we cannot reject the null hypothesis that they are, indeed, Gaussian.

| Noise | analytical $\varepsilon$ | $\varepsilon_{\text{lo}}$-all | $\varepsilon_{\text{est}}$-all | $\varepsilon_{\text{lo}}$-final | $\varepsilon_{\text{est}}$-final |
|---|---|---|---|---|---|
| 0.0 | $\infty$ | 6.25 | 48300 | 3.86 | 5.72 |
| 0.16 | 33.8 | 2.87 | 17.9 | 1.01 | 1.20 |
| 0.18 | 28.1 | 2.32 | 12.0 | 0.788 | 1.15 |
| 0.195 | 24.8 | 2.02 | 8.88 | 0.723 | 1.08 |
| 0.25 | 16.9 | 0.896 | 3.86 | 0.550 | 0.818 |
| 0.315 | 12.0 | 0.315 | 1.50 | 0.216 | 0.737 |

Table 5: Comparing $\varepsilon$ estimates using all model deltas vs. using the final model only. $\varepsilon_{\text{lo}}$ is the empirical 95% lower bound from our modified Jagielski et al. [2020] method. The high values of $\varepsilon_{\text{est}}$-all indicate that membership inference is easy when the attacker has access to all iterates. On the other hand, when only the final model is observed, $\varepsilon_{\text{est}}$-final is far lower.

with learning rates 0.031 and 1.0 respectively, and momentum of 0.9 on the server. The client batch size is 16.

Table 5 shows the empirical epsilon estimates using either all model iterates or only the final model. As with Stackoverflow next word prediction, using all iterates and a low amount of noise gives us estimates close to the analytical upper bound, while using only the final model gives a much smaller estimate.

Figure 3 demonstrates the effect of increasing the number of canary repetitions for EMNIST. The results are qualitatively similar to the case of Stackoverflow.

## G  EMPIRICAL COMPARISON WITH CANIFE

As discussed in the main text, our method is significantly more general than the CANIFE method of Maddock et al. (2022). CANIFE periodically audits individual rounds to get a per-round $\hat{\varepsilon}_r$, estimates the noise for the round $\hat{\sigma}_r$ by inverting the computation of $\varepsilon$ for the Gaussian mechanism, and uses

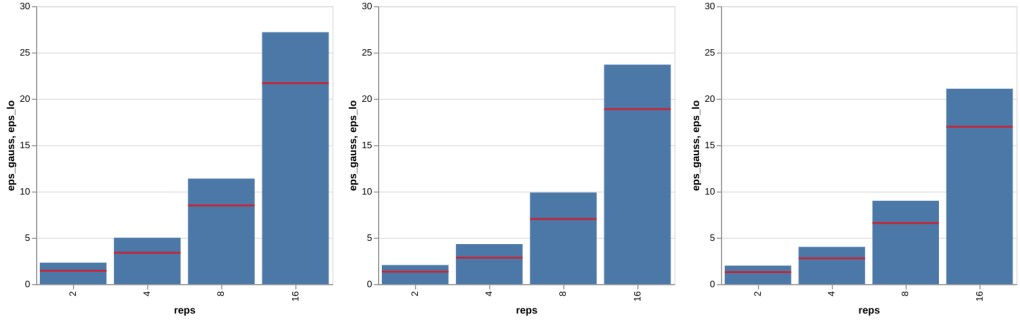

Figure 3: Blue bars are our $\varepsilon_{\text{est}}$ and red ticks are the $\varepsilon_{\text{lo}}$ 95%-confidence lower bound for three noise multipliers (0.16, 0.18, 0.195) and four numbers of canary repetitions. Our estimate of epsilon increases sharply with the number of canary repetitions, confirming that limiting client participation improves privacy.

| CANIFE | $\varepsilon_{\text{lo}}$ | ours | analytical $\varepsilon$ |
|---|---|---|---|
| $0.88 \pm 0.12$ | $1.82 \pm 0.46$ | $6.8 \pm 1.1$ | 34.5 |

Table 6: Mean and standard deviation over 50 runs of three $\varepsilon$ estimates, and analytical $\varepsilon$ bound.

standard composition theorems to determine a final cumulative epsilon. Therefore the estimate will be inaccurate if the assumptions of those composition theorems do not strictly hold, for example, if clients are not sampled uniformly and independently at each round, or the noise is not in fact isotropic Gaussian and independent across rounds. In contrast, our method will detect if any unanticipated aspect of training causes some canaries to have more influence on the final model in terms of higher canary/model dot product, leading in turn to higher $\varepsilon$ estimates. Also CANIFE's method of crafting canaries is model/dataset specific, and computationally expensive.

It is still interesting to see how the methods compare in the limited setting where CANIFE's assumptions do hold. Unfortunately a comparison of two "estimates" is not straightforward when there is no solid ground truth. In this case, we do not have any way to determine the "true" $\varepsilon$, even inefficiently, because doing so would require designing a provably optimal attack. However, we can use a strong attack to determine a lower bound on $\varepsilon$ directly from Eq. (1). If the lower bound exceeds CANIFE's $\varepsilon$, then we know it is an underestimate.

We trained a two-layer feedforward network on the fashion MNIST dataset. Following experiments in Maddock et al. (2022), we used a canary design pool size of 512, took 2500 canary optimization steps to find a canary example optimizing pixels and soft label, ran auditing every 100 rounds with 100 attack scores on each auditing round. We trained with a clip norm of 1.0 and noise multiplier of 0.2 for one epoch with a batch size of 128, which corresponds to an analytical $\varepsilon$ of 34.5.

To compute a lower bound, we trained with 1000 inserted gradient canaries sampled from the unit sphere. We computed attack statistics by taking the maximum over steps of the cosine of the canary to the model delta for the step, just as in Algorithm 3. Only instead of fitting them to a Gaussian to produce an $\varepsilon$ estimate, we compute a lower bound on $\varepsilon$ by bounding the FPR and FNR of the attack as in Jagielski et al. Following Zanella-Béguelin et al. (2023) we use the tighter and more centered Jeffreys confidence interval.

The results are shown in Table 6. CANIFE's $\varepsilon$ of 0.88 is somewhat lower than the lower bound of 1.82, while ours is significantly higher, at 6.8. We still cannot rule out that our method is not overestimating $\varepsilon$, but at least we can say that it is in the plausible (but wide) range $[1.82, 34.5]$ while CANIFE's is not.

There are several reasons why CANIFE might be underestimating $\varepsilon$. Although the authors bill the method as a "measurement" of $\varepsilon$, not a lower bound, nevertheless, it uses a lower bound on the per-round $\varepsilon_r$ to estimate the per-round $\sigma_r$. Thus, it has a built-in bias toward lower $\varepsilon$, and there is a maximum $\varepsilon$ it cannot exceed simply due to the finite sample size. CANIFE also searches for a canary example whose gradient is nearly orthogonal to a held-out set of data (not necessarily to the actual batch being audited). Our method uses a random gradient which is provably nearly orthogonal to the other gradients in the batch with high probability, leading to a stronger attack.

## H  COMPARISON OF ASSUMPTIONS AND REQUIREMENTS OF EMPIRICAL PRIVACY ESTIMATION METHODS

As discussed in Section 2, related work in privacy auditing/estimation relies on various assumptions on the auditor's knowledge and capability. Here we summarize the major differences.

The standard assumption in auditing centralized private training algorithm is a black-box setting where the auditor only gets to control the training data and observes the final model output. In practice, many private training algorithms guarantee privacy under releasing all the intermediate model checkpoints. One can hope to improve the estimate of privacy by using those checkpoints as in (Jagielski et al., 2023). If the auditor can use information about how the minibatch sequence is drawn and the distribution of the privacy noise, which is equivalent to assuming that the auditor controls the privacy noise and the minibatch, one can further improve the estimates.

| | | auditor controls | auditor receives |
|---|---|---|---|
| **Central** | Jagielski et al. (2020) | train data | final model |
| | Zanella-Béguelin et al. (2023) | train data | final model |
| | Pillutla et al. (2024) | train data | final model |
| | Steinke et al. (2024) | train data | final model |
| | Jagielski et al. (2023) | train data | intermediate models |
| | Nasr et al. (2023) | train data, privacy noise, minibatch | intermediate models |
| **FL** | Algorithm 2 | client model update | final model |
| | Algorithm 3 | client model update | intermediate models |
| | CANIFE (Maddock et al., 2022) | client sample, privacy noise, minibatch | intermediate models |

Table 7: Assumptions of different auditing approaches from the literature. For each paper, we state the most relaxed condition the technique can be applied to, since they can be generalized in straightforward manner to scenarios with more strict assumptions on the auditor's control and observation. Within each category of {central training, federated learning}, the lists are ordered from least to most strict assumptions.

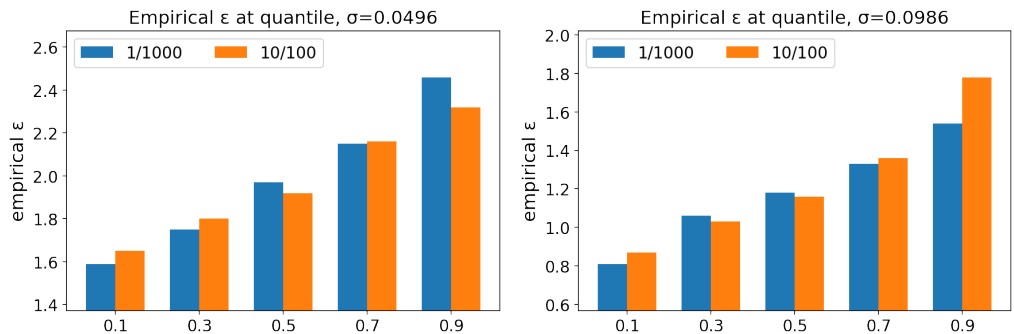

Figure 4: Quantiles of $\hat{\varepsilon}$ over fifty experiments using either one run with 1000 canaries or ten runs with 100 canaries each. For both noise multipliers, the distributions are very close.

In the federated learning scenario, we assume a canary client can return any model update. Note that while CANIFE only controls the sample of the canary client and not the model update directly, CANIFE utilises the Renyi-DP accountant with Poisson subsampling implemented via the Opacus library, which is equivalent to the auditor fully controlling the sequence of minibatches (cohorts in FL terminology). Further, the privacy noise is assumed to be independent spherical Gaussian, which is equivalent to the auditor fully controlling the noise.

Table 7 compares the assumptions of different auditing approaches from the literature.

## I  EXPERIMENTS COMPARING WHEN MULTIPLE RUNS ARE USED

In the limit of high model dimensionality, canaries are essentially mutually orthogonal, and therefore they will interfere minimally with each other's cosines to the model. In this section we give evidence that even in the range of model dimensionalities explored in our experiments, including many canaries in one run does not significantly perturb the estimated epsilon values. Ideally we would train 1000 models each with one canary to collect a set of truly independent statistics. However this is infeasible, particularly if we want to perform the entire process multiple times to obtain confidence intervals. Instead, we reduce the number of canaries per run by a factor of ten and train ten independent models to collect a total of 1000 canary cosine statistics from which to estimate $\varepsilon$. We repeated the experiment 50 times for two different noise multipliers, which still amounts to training a total of 1000 models. (Ten runs, two settings, fifty repetitions.)

The results on the stackoverflow dataset with the same setup as in Section 5 are shown in Figure 4. We report the 0.1, 0.3, 0.5, 0.7 and 0.9 quantile of the distribution of $\hat{\varepsilon}$ over 50 experiments. For both noise multipliers, the distributions are quite close. Our epsilon estimates do not seem to vary significantly even as the number of canaries per run is changed by an order of magnitude.

