# OpenReview forum: "One-shot Empirical Privacy Estimation for Federated Learning"
_ICLR.cc/2024/Conference — ICLR 2024 oral_

### Official Review · Reviewer_ZyjW · 2023-10-17

**Soundness:** 4 excellent
**Presentation:** 3 good
**Contribution:** 3 good
**Rating:** 8
**Confidence:** 4

**Summary:**

The paper proposes a new method to empirically estimate the privacy loss of an algorithm, targeted at federated learning. The method can estimate the privacy loss from just one run of the algorithm, which makes it easy to apply. This is done by randomly sampling canary vectors, which are presented to the central aggregator as regular updates. The cosine similarity between each of the canaries and the output of the audited algorithm is then used as a test statistic to infer whether the canary was in the training set or not. The test statistic's distribution, both with or without the canaries, can be approximated with simple Gaussians, as the canaries are likely to be orthogonal to each other and the actual update when the space is high-dimensional. The Gaussian approximations are justified with asymptotic theorems, and the whole method is evaluated on several experiments.

**Strengths:**

The paper is very well-written and easy to understand. The proposed method fairly simple, so it should be easy to implement, and potentially to extend to other settings. The main idea of randomly sampling canaries that are orthogonal with everything else with high probability is novel to my knowledge. Being able to estimate the privacy loss in a single training run, with minimal effect on the target model's accuracy, makes the method fairly practical.

**Weaknesses:**

The analytical privacy bound doesn't seem like a good "ground truth" for the comparison with CANIFE, as the analytical bound is expected to be much larger than necessary. The best $\epsilon$ to return would be the upper bound that any membership inference attack could achieve, which is of course not known. It is possible that your method is simply overestimating the best $\epsilon$, and is closer to the analytical because of that, and not because it is better at estimating the best $\epsilon$ than CANIFE. This does not seem a too remote of a possibility, as if your result was accurate, it would imply that CANIFE grossly underestimated the $\epsilon$, and simply doesn't work in the setting.

A better "ground truth" would be an $\epsilon$ lower bound obtained from the TPR and FPR of some strong membership inference attack. It might be possible to use your method as this attack by thresholding the cosine test statistics $g_i$, and estimating the TPRs and FPRs that different thresholds give empirically.

The proofs are missing some details, which makes them harder to understand and check than necessary. In the proof of Theorem 3.1, what is the value of the measure $A_d(\theta)$, and how is it derived from the $(d-2)$ measure of its boundary? In the proof of Theorem 3.2, $t$ is not defined. You should also name the theorem that allows you to conclude convergence in distribution from pointwise convergence of the density function, and you should explicitly account for the fact that the density of $t$ is 0 outside $[-\sqrt{d}, \sqrt{d}]$.

These two issues are the main reason for my score, and I will increase the score if they are addressed.

Regarding the challenge to come up with an attack that breaks your method, I can come up with two scenarios where this is likely to happen. The first is not using a cryptographically secure random number generator to generate the Gaussian noise, which would allow an attacker to remove the noise if they can break the insecure RNG. The second is an attack based on finite-precision issues with floating point numbers (see Mironov 2012 and Holohan and Braghin 2021), if the noise is not sampled with defenses against these in place. I don't think your method would detect these, as it is only looking at the noise as a real number, and detecting either scenario seems to require looking at the noise as a finite-precision float. Of course, your method is not designed to detect anything like these in the first place, and I don't think any of alternatives are either, so this is not a large limitation, but it should still be mentioned.

References:
- N. Holohan, S. Braghin "Secure random sampling in differential privacy" Computer Security – ESORICS 2021
- I. Mironov "On significance of the least significant bits for differential privacy" ACM Conference on Computer and Communications Security 2012

Minor comments:
- The Anderson-Darling test should be cited.
- The assumption that $n = o(d)$ should be stated in Theorem 3.3, not just as a footnote.
- Font size in Figures 1 and 2 are too small.
- Table 6 would be much easier to read as a plot of the quantiles, which would also allow showing much more than 5 quantiles.
- Specifying the exact CNN and LSTM architectures in the experiments would be good for reproducibility, as the code is not included in the submission.
- Using \left and \right on the curly braces in Equation (1) would make the equation easier to read.

Comments on references:
- Feldman et al. (2021) URL points to arXiv, not the conference submission
- Capitalisation in some paper titles, for examples "rényi" in Feldman et al. (2023)
- arXiv papers have inconsistent format, for example compare Maddock et al. (2022) and Pillutla et al. (2023)
- Steinke (2022) is missing the publication forum
- Zanella-Beguelin et al. (2022) and (2023) are the same paper

**Questions:**

- Is the distribution of the test statistic in Theorem 3.3 (asymptotically) Gaussian?
- What are the upper bounds for the $\epsilon$ confidence intervals in Table 2? It looks like the intervals are huge in the -all columns.
- Which plots correspond to which canary levels in Figure 1 and which epsilons in Figure 2? Adding the canary levels and epsilons to the plots, for example in subplot titles, would make the figures much easier to read.
- Is it possible to use your method to audit standard DP-SGD? If so, how does the method compare to other auditing methods? For example, do the other methods require more than one training run?

---

> ### Author Response · Authors · 2023-11-13
>
> Thank you for the careful review. You make some excellent points about the difficulty of comparing two empirical privacy estimates, which we will address in a subsequent comment. In the interest of getting the conversation started during this short review period, we will address the rest of your comments first. You said your other main concern and reason for your rating is regarding the proofs. Please let us elucidate that for you here.
>
> In Theorem 3.1, we could give a formula for the value of the measure of the spherical cap $A_d(\theta)$, but it is actually not necessary. Since we are interested in the rate of change of $A_d(\theta)$ with respect to $\theta$, all we need is the measure of its boundary: as $\theta$ increases, $A_d(\theta)$ increases by an amount that is proportional to the measure of its boundary, which is a $(d-1)$-sphere of radius $\sin \theta$. The measure of a hypersphere of radius $R$ can be found, for example, here https://zhangyk8.github.io/teaching/file/Exercise_4_insight.pdf. We can clarify and add a footnote to the proof citing this reference.
>
> We apologize for the difficulty in following the proof of Theorem 3.2 and we appreciate your suggestions to clarify it. We will clearly define $\hat{f}_d(t)$ to be the density function of $\tau \sqrt{d}$ (where $\tau$ is defined in Theorem 3.1). We will properly cite Scheffe’s Theorem, which is the result that pointwise convergence of the density function implies convergence in distribution. As for your other comment, strictly speaking, the density of $\tau \sqrt{d}$ is defined on all of $\mathbb{R}$, but it is equal to 0 outside of $-[\sqrt{d}, \sqrt{d}]$. Since $\sqrt{d} \rightarrow \infty$ as $d$ increases, the pointwise limit holds for all $t \in \mathbb{R}$. We will amend the formula for $\hat{f}_d(t)$ to show how it is defined on all of $\mathbb{R}$.
>
> We hope that completely addresses your concerns about the proofs.
>
> Other questions:
> * *Is the distribution of the test statistic in Theorem 3.3 (asymptotically) Gaussian?* We noticed just now that we misstated in the paper at the bottom of page 4 that the distribution of the test statistic is Gaussian. We apologize for that mistake and we will correct it. In fact, there are some tricky subtleties here: since the test statistics are not independent, we need to make some statement about their joint distribution, for example, that the joint distribution approaches the product of independent Gaussians. While it would be surprising if that were not the case, we don’t currently have a proof. All of that said, we would like to emphasize that it is not necessary to completely characterize their distribution in order to prove the main result of Section 3 (asymptotic correctness of Algorithm 1). Algorithm 1 depends only on the empirical mean and variance of the canary cosines, which Theorem 3.3 proves are asymptotically correct.
> * *What are the upper bounds for the confidence intervals in Table 2? It looks like the intervals are huge in the -all columns.* As discussed in the second paragraph of Sec. 5, $\varepsilon_\text{lo}$ is not strictly speaking a lower confidence bound on our $\varepsilon_\text{est}$. It is a modified version of the Jagielski et al. method. This method provides a lower bound on $\varepsilon$, but is statistically limited by the number of samples, so that it cannot give very high values. In this case it cannot go higher than 6.24, where the empirical false negative rate is 0. We include it to show that a) our method does not suffer from this limitation, and b) even when the value is below 6.24, our method is not falsified by the lower bound.
> * *Which plots correspond to which canary levels in Figure 1 and which epsilons in Figure 2?* The subplots in Figs 1 and 2 correspond 1:1 from left to right, but we agree this will be clearer if we add epsilon labels to the subplots, which we will do.
> * *Is it possible to use your method to audit standard DP-SGD? If so, how does the method compare to other auditing methods? For example, do the other methods require more than one training run?* Certainly, there is nothing about our method that would prevent it from being used for DP-SGD (we allude to this in the first footnote). It would have the same advantages relative to most other methods (which indeed generally require more than one run). Our method of generating a nearly-mutually-orthogonal set of gradient canaries can also be combined with the recent technique of Steinke et al. to obtain a formal lower bound on DP-SGD’s epsilon in one training run.
> * *Regarding the challenge to find an attack that breaks your method…* You are correct that our method (like all work in empirical privacy auditing) would not detect breaches of differential privacy from attacks exploiting not using a cryptographically secure source of randomness, or the finite precision of floating point numbers. We will be more explicit in the conclusion that our method is not designed to detect such attacks.

---

> ### Comment · Reviewer_ZyjW · 2023-11-15
>
> Thank you for the reply. You addressed many of my concerns and questions, but I still have a few remaining ones.
>
> > In Theorem 3.1, we could give a formula for the value of the measure of the spherical cap $A_d(\theta)$, but it is actually not necessary. Since we are interested in the rate of change of $A_d(\theta)$ with respect to $\theta$, all we need is the measure of its boundary: as $\theta$ increases, $A_d(\theta)$ increases by an amount that is proportional to the measure of its boundary, which is a $(d-1)$-sphere of radius $\sin \theta$.
>
> ~~I still don't fully follow this argument. I see that when $A_d$ and $M_d$ are written in terms of the radius $r$, $A_d(r) = C_1r^{d-1}$ and $M_d(r) = C_2r^{d-2}$, so $\frac{d}{dr}A_d(r) = (d-1)C_1 r^{d-2} \propto M_d(r)$, like you wrote. However, the derivative in the proof is with respect to $\theta$, so $A_d(\theta) = C_1\sin^{d-1} \theta$ and $\frac{d}{d\theta} A_d(\theta) = (d-1)C_1\sin^{d-2}\theta \cdot \cos \theta$. Where does the extra $\cos\theta$ factor go?~~
>
> After thinking about this a bit more, I think I understand the idea. Is the idea that $A_d(\theta) = \int_0^\theta M_d(\phi)d\phi$, so $\frac{d}{d\theta}A_d(\theta) = M_d(\theta)$?
>
> Now that I'm thinking about this proof again, I think the claim "the boundary of $A_d(\theta)$ is a $(d-1)$-dimensional sphere with radius $\sin \theta$" should be proven explicitly. The claim is clear when $d = 3$, but in higher dimensions it is not trivial to prove.
>
> > All of that said, we would like to emphasize that it is not necessary to completely characterize their distribution in order to prove the main result of Section 3 (asymptotic correctness of Algorithm 1). Algorithm 1 depends only on the empirical mean and variance of the canary cosines, which Theorem 3.3 proves are asymptotically correct.
>
> Why is it not necessary for the asymptotic distribution to be Gaussian? The $\epsilon$ estimation method in Algorithm 1 depends on the distribution of the canary cosines, not just their mean and variance. If the asymptotic distribution is not Gaussian, the $\epsilon$ estimation method should be changed to reflect that.

---

> > ### Author Response · Authors · 2023-11-16
> >
> > Thank you for the reply.
> >
> > * Your statement about the measure of the spherical cap being the integral of the measure of its boundary with respect to $\theta$ is spot on. We will add that note to the proof.
> > * *The claim "the boundary of the spherical cap is a $(d-1)$-dimensional sphere with radius $\sin \theta$ should be proven explicitly. The claim is clear when $d=3$, but in higher dimensions it is not trivial to prove.* The boundary of the spherical cap is the set of points on the sphere with angle to the reference vector $v$ equal to $\theta$. Without loss of generality, let the reference vector $v$ be $(1,0, \ldots, 0)$. Having angle $\theta$ to $v$ means $x_1 = \cos \theta$, so we are interested in the set of points $(\cos \theta, x_2, \ldots, x_d)$ where $\cos^2 \theta + \sum_{i=2}^d x_i^2= 1$. The distance from such a point to the point $(\cos \theta, 0, \ldots, 0)$ is precisely $\sqrt{\sum_{i=2}^d x_i^2} = \sqrt{1 - \cos^2 \theta} = \sin \theta$. So they are a hypersphere in $(d-1)$ dimensions ($d-1$ free coordinates) with center $(\cos \theta, 0, \ldots, 0)$ and radius $\sin \theta$. Once again we really appreciate your suggestion to clarify this! We will add it to the proof which should help future readers.
> > * *Why is it not necessary for the asymptotic distribution to be Gaussian?* The method of Algorithm 1 is to generate a set of samples, fit them to a Gaussian, and compare that Gaussian to the null hypothesis distribution. Please see line 10 of Algorithm 1, which specifies that we compare (using the exact computation of $\varepsilon$ from two Gaussian distributions described in Appendix E) the distributions $\mathcal{N}(0, 1/d)$ and $\mathcal{N}(\hat{\mu}, \hat{\sigma}^2)$, where $\hat{\mu}$ and $\hat{\sigma}^2$ have been estimated from the samples. In any case, all we have is a set of samples; we do not have access to the underlying distribution. Now the question is: why is fitting them to a Gaussian the right thing to do? And there are two reasonable answers. 1) Because the samples really are (asymptotically, independently) Gaussian distributed. 2) Because by doing so, we recover the correct $\varepsilon$. Our intuition is that 1) is true. We have experimental evidence by looking at histograms of the samples. We agree that it would be odd if the samples were not Gaussian distributed, but fitting a Gaussian to them somehow coincidentally produced the correct $\varepsilon$. However, proving 1) is not necessary for the algorithm to be correct; proving 2) is sufficient, and that is what we have done. Please let us know if you would like any additional clarification on this front.
> >
> > We will reply soon about the CANIFE comparison.

---

> > > ### Comment · Reviewer_ZyjW · 2023-11-16
> > >
> > > The proof about the shape of the spherical cap is very good. I still have a few questions and suggestions on the other points.
> > >
> > > > Your statement about the measure of the spherical cap being the integral of the measure of its boundary with respect to is spot on. We will add that note to the proof.
> > >
> > > You should also add a justification for $A_d(\theta) = \int_0^\theta M_d(\phi)d\phi$. I was able to verify that for $d = 3$ using by considering the spherical cap a surface of revolution, but I don't know if that generalises to higher dimensions.
> > >
> > > > However, proving 1) is not necessary for the algorithm to be correct; proving 2) is sufficient, and that is what we have done.
> > >
> > > Can you point out where in the paper the proof of 2) is? I agree that the test statistics are likely asymptotically Gaussian, as your results in Appendix B show empirically, but a mathematical proof would be much more convincing.
> > >
> > > The test statistics are normalised dot products, meaning they are sums over several random variables. Would it be possible to use one of the CLT generalisations that works for sums of non-i.i.d. random variables to show that the test statistics are asymptotically Gaussian?

---

> ### Author Response · Authors · 2023-11-16
> **Regarding "ground truth" for empirical comparison**
>
> Now we would like to address your concern in the original review about the comparison to CANIFE. We absolutely concur that it is difficult to specify a good “ground truth” for comparison. We do not have any way to determine the “true” $\varepsilon$, even inefficiently, because doing so would require designing a provably optimal attack.
>
> But what you suggest is perhaps the best we can do: use a strong attack to determine a lower bound on $\varepsilon_\text{lo}$. If the lower bound exceeds CANIFE's $\varepsilon$, then we know CANIFE is underestimating $\varepsilon$. We still cannot rule out that our method is not overestimating $\varepsilon$, but at least it has not been invalidated by the lower bound.
>
> The CANIFE authors bill the method as a “measurement” of $\varepsilon$, not a lower bound. Nevertheless, CANIFE uses a lower bound on the per-round $\varepsilon_r$ to estimate the per-round $\sigma_r$. Thus, it has a built-in bias toward lower $\varepsilon$, and there is a maximum $\varepsilon$ it cannot exceed simply due to the finite sample size. CANIFE also searches for a canary example whose gradient is nearly orthogonal to a held-out set of data (*not* necessarily to the actual batch being audited). Our method uses a random gradient which is provably nearly orthogonal to the other gradients in the batch with high probability, leading to a stronger attack.
>
> So we ran the experiment you suggested. We inserted our random gradient canaries, thresholded on the cosine test statistics, and computed a $\varepsilon$ lower bound from the TPR and FPR of the optimal threshold. The result was a lower bound of $\varepsilon = 1.82 \pm 0.46$. This is indeed significantly higher than CANIFE’s $0.88 \pm 0.12$. It would be a stronger validation of our method if we could produce a lower bound that is closer to our value of $6.8 \pm 1.1$, but at least we can say that our estimate is in the plausible (but wide) range $(1.82, 34.5)$ while CANIFE’s is not. (34.5 is the analytical upper bound.) We will add this result and discussion to the CANIFE comparison in Appendix F.

---

> ### Author Response · Authors · 2023-11-16
>
> *You should add a justification for $A_d(\theta) = \int_0^\theta M_d(\phi) d\phi$.* The measure of the volume element of the spherical cap at angle $\phi$ and of width $d \phi$ is just $M_d(\phi) d\phi$. Integrate them to get the measure of the whole cap. It’s not considering the cap as a surface of revolution, but rather integrating over the angle $\theta$, as is done in [this comment](https://math.stackexchange.com/q/3926922).
>
> *Can you point out where in the paper the proof of 2) is?* It is the consequence of Theorems 3.1-3, as argued in the paragraph at the bottom of page 4 (modulo the error in that paragraph we mentioned in our first response). However, based on this discussion, it has become clear to us that the way we wrote it there is potentially confusing. Here is a clearer exposition.
>
> Line 10 of Algorithm 1 says that we compute $\hat{\varepsilon} \leftarrow \varepsilon(\mathcal{N}(0, 1/d)\ ||\ \mathcal{N}(\hat{\mu}, \hat{\sigma}^2); \delta)$, where $\hat{\mu}$ and $\hat{\sigma}^2$ have been estimated from the samples, and we derive the exact computation of $\varepsilon$ when $A(D)$ and $A(D’)$ are two arbitrary Gaussian distributions in Appendix E. This can be written equivalently as $\hat{\varepsilon} \leftarrow \varepsilon(\mathcal{N}(0, 1)\ ||\ \mathcal{N}(\sqrt{d} \hat{\mu}, d \hat{\sigma}^2); \delta)$, because it is just a scaling of both distributions by a factor of $1/\sqrt{d}$, which does not change the $\varepsilon$. Note that the function $\varepsilon$ depends only on $\hat{\mu}$ and $\hat{\sigma}^2$ -- nothing else about the samples. Now Theorem 3.3 demonstrates that $\sqrt{d}\hat{\mu} \overset{p}{\longrightarrow} 1/\sigma$ and $d\hat{\sigma}^2 \overset{p}{\longrightarrow} 1$. Therefore by the Mann-Wald theorem, the estimate converges in probability to $\varepsilon (\mathcal{N}(0, 1)\ ||\ \mathcal{N}(1 / \sigma, 1); \delta)$. Now these two distributions are just a scaling of $A(D) \sim \mathcal{N}(0, \sigma^2)$ and $A(D') \sim \mathcal{N}(1, \sigma^2)$ by a factor of $1 / \sigma$, so the $\varepsilon$ is the same as the mechanism we are auditing (see the first paragraph of Sec. 3). This proves our claim. We will rewrite the last paragraph on page 4 to include this argument.
>
> We emphasize again that it is not essential that the cosines $g_i$ actually *be* Gaussian distributed for this argument to hold. Adding a proof that they are truly Gaussian might aid our intuition, but it is not necessary if what we are most concerned about is correctness of the algorithm– that it outputs the correct $\varepsilon$. Also we want to be clear that it would not be sufficient to show that each cosine sample is marginally Gaussian (which would be relatively easy to show). To see this, note that If we proved only that much, it could pathologically be the case that all of the samples are always identical to each other on each run (but Gaussian distributed across runs) so that the empirical variance on any run is zero. Then our method would be broken. But we proved in Theorem 3.3 that the empirical variance approaches $1/d$.

---

> > ### Comment · Reviewer_ZyjW · 2023-11-17
> >
> > Thanks for running the experiment. I fully agree with your interpretation of the result, which clearly shows that your method is more reliable than CANIFE.
> >
> > > The measure of the volume element of the spherical cap at angle $\phi$ and of width $d\phi$ is just $M_d(\phi)d\phi$.
> >
> > I understand the intuition behind this in 3 dimensions. I agree that it probably is true in higher dimensions, but that should still be proven. I'm also not fully convinced by these kinds of arguments that rely on infinitesimal elements. They can be useful for intuition and can often be translated into proofs, but they can also go wrong in ways that are not immediately apparent. I would expect this to be even more likely when the infinitesimal elements are arbitrary-dimensional.
> >
> > > It is the consequence of Theorems 3.1-3, as argued in the paragraph at the bottom of page 4 (modulo the error in that paragraph we mentioned in our first response). However, based on this discussion, it has become clear to us that the way we wrote it there is potentially confusing. Here is a clearer exposition.
> >
> > Thanks for the explanation, now I understand how this argument works. It might even be useful to make the explanation a theorem, so that the important conclusion is clear.

---

> > > ### Author Response · Authors · 2023-11-17
> > >
> > > * *It might even be useful to make the explanation a theorem, so that the important conclusion is clear.* Yes, we will do that.
> > > * Regarding the integral to determine the $(d-1)$-measure of the hyperspherical cap. We found a proper reference that uses this exact result. Hopefully this should alleviate your concerns. Please see the first equation in the section “Area of a hyperspherical cap” of [this paper](https://docsdrive.com/pdfs/ansinet/ajms/2011/66-70.pdf). We will cite this work in our proof.
> > >
> > > Once again, we really appreciate your feedback which should make the submission more rigorous! We hope we have now addressed all of the questions and concerns you mentioned in the original review.

---

> > > > ### Comment · Reviewer_ZyjW · 2023-11-17
> > > >
> > > > Thanks, this addresses the last of my major concerns. I'll increase my score accordingly.

---

### Official Review · Reviewer_zCeQ · 2023-11-01

**Soundness:** 3 good
**Presentation:** 3 good
**Contribution:** 2 fair
**Rating:** 8
**Confidence:** 3

**Summary:**

This work proposes a one-shot empirical privacy estimation method for DP-FedAvg. Instead of choosing canary examples, the authors choose canary clients that releases randomly generated updates to the system. The idea is then to measure the cosine similarity between the overall updates and random updates to determine the effect of canary clients in the overall system. Then using the similarity they determine the empirical privacy using the final model.

**Strengths:**

- The paper is well-written and easy to follow. Introduction and motivation of the method is clearly communicated.
- Consideration of estimation of $\epsilon$ for Gaussian mechanism is spot on, since in that case authors are able to prove that the estimation becomes correct asymptotically (in d).
- Experiments with multiple datasets and architectures are presented.

**Weaknesses:**

- Using canary clients seems more inefficient compared to using canary examples. More resource allocation might be needed.
- In the paper it is suggested that
"In production settings, a simple and effective strategy would be to designate a small fraction of real clients to have their model updates replaced with the canary update whenever they participate."
but this would destroy the representation of such clients and problematic, especially, in data heterogenous settings. It would also result in fairness problems.
- It is hard to interpret the provided empirical comparison with CANIFE. Other than the assumptions for CANIFE, it is not clear to me how this method is better.
- The authors claim that their method is agnostic to architecture knowledge, but aren't the $c_j$ are of same dimension as the architecture? Hence would not designing such canaries would require architecture knowledge?

**Questions:**

- I could not see any dependence on canary clients- true clients ratio in your results. Why is there not a dependence on $k/m$ in Theorem 1-3? What are the effects of this ratio empirically and theoretically?
- In Table 2 what is the goal of comparing to $\epsilon_{lo}$, and what is the conclusion?
- I think for a setting in experiments you should also vary the dimensionality of the model and obtain a empirical privacy-dimensionality relationship (by keeping analytical one constant).
- What is the conclusion of Appendix F? Even if your method is close to the analytical method, do you think that is enough evidence to say that it is a better method? I'm curious if there are any other ways to make a comparison between two methods (such as using Canife in the warm-up experiment).
- How could one extend this method to other privacy mechanisms other than Gaussian?

---

> ### Author Response · Authors · 2023-11-14
>
> We appreciate your careful review and suggestions to improve our paper. We believe we can answer most of your concerns.
>
> * *Using canary clients seems more inefficient compared to using canary examples. More resource allocation might be needed.* The reason to use canary clients, as opposed to canary examples, is that we are estimating user-level DP. We need to measure the impact on the model of adding or removing an entire client. Using canary examples would only give an estimate of example-level DP. In any case, resource allocation should not be an issue; if anything it requires fewer resources to forego training on some client’s data and replace its entire update with a canary, vs. replacing one of its example gradients with a canary (of the same dimensionality) and performing local training.
> * *Wouldn’t replacing some users’ data destroy their representation and result in fairness problems?* It’s true that a small fraction of users’ data would be lost. In our stackoverflow dataset there were 341k users. In the production DP-FL-trained language models of Xu et al. (2023), there are more than 3M users for almost all of the languages. If we replaced 1k of those users with canaries, we would lose 0.033% of the data. Because the users would be sampled uniformly at random from the population (for example, by hashing on a user ID), it seems to us there should not be a fairness problem, even in the data heterogenous setting. If there is a definition of fairness that would be violated, we would be interested in hearing about it so we could discuss a different strategy (e.g., augmenting the dataset with canary users, rather than swapping real users with canary users). The advantage of replacing users’ data, as stated in the paper, is that it guarantees that the canary client participation is exactly according to the natural distribution.
> * *It is hard to interpret the provided empirical comparison with CANIFE.* We agree, the primary advantages of our method with respect to CANIFE are that it makes fewer assumptions and is easier to implement. Although CANIFE is billed as an empirical privacy “measurement”, not a lower bound, it uses a lower bound to audit individual rounds. We believe this causes the estimates it produces to be on the low side. The experiment shows that our estimate is closer to the analytical upper bound. Also, CANIFE's estimate is lower because it attempts to craft a canary example whose gradient is nearly orthogonal to the gradients of a small auxiliary set of examples. In contrast, we use canary updates that are provably nearly orthogonal to the true updates, leading to a stronger attack.
> * *The authors claim that their method is agnostic to architecture knowledge, but aren't the $c_j$ of the same dimension as the architecture?* Yes, the dimensionality of the model architecture must be known. Being agnostic to all other aspects of model architecture means that it is easy to add an empirical privacy estimation module to an existing FL system: when the system trains a model, all the EPE module needs to do is insert random canary vectors during training (which, of course, must be of the appropriate dimensionality). This is in contrast to CANIFE, for example, which requires an architecture-specific method and in-distribution auxiliary training data to design canary examples, which is completely different for continuous or discrete data.
> * *Why is there not a dependence on $k/n$ in Theorem 1-3?* (You wrote $m$, but we assume you meant $n$. Actually this is a good reminder that it is confusing that we used $m$ in section 5. We will unify the notation. Thank you.) Theorems 3.1-3 deal with the case where $d$ becomes large. So long as $d$ asymptotically dominates both $k$ and $n$, the ratio $k/n$ should not matter. In high dimensions, the canaries become orthogonal to all real data vectors and to each other, no matter how many there are.
> * *In Table 2 what is the goal of comparing to $\varepsilon_\text{lo}$, and what is the conclusion?* As discussed in the second paragraph of Sec. 5, the modified Jagielski et al. method in the table provides a lower bound on the true $\varepsilon$, but is statistically limited by the number of samples, so that it cannot give very high values. In this case it cannot go higher than 6.24, where the empirical false negative rate is 0. We include it to show that a) our method does not suffer from this limitation, and b) even when the value is below 6.24, our method is not falsified by the lower bound. We will try to make the reason for this comparison more clear in the paper.
>
> Z. Xu, Y. Zhang, G. Andrew, C. A. Choquette-Choo, P. Kairouz, H. Brendan McMahan, J. Rosenstock, Y. Zhang (2023). Federated Learning of GBoard Language Models with Differential Privacy. https://arxiv.org/abs/2305.18465

---

> ### Author Response · Authors · 2023-11-14
>
> * *I think for a setting in experiments you should also vary the dimensionality of the model and obtain a empirical privacy-dimensionality relationship.* We proved in Section 3 that so long as the dimensionality is large enough the epsilon estimate does not depend on $d$. However, to alleviate your concern, we reran the experiment of section 3 with $d$ varying from 10k to 10M. Please see our second comment to reviewer LDHu for those results, which we will add to the paper. But we emphasize that our method is not really designed for such small models.
> * *What is the conclusion of Appendix F?* We discussed this in the third bullet above.
> * *How could one extend this method to other privacy mechanisms other than Gaussian?* This is a very interesting question. Probably the canary distribution would have to change. For example, for the Laplace mechanism maybe the canaries could be sampled from the set of vectors of unit L1 norm. But to be honest we have not thought much about this, since the Gaussian mechanism is the building block of DP-FedAvg and its variants. We will mention this as an interesting open question in our conclusion.
>
> Have we answered all of your questions? Do you have any other concerns we could address to merit a higher score? Thank you.

---

> ### Author Response · Authors · 2023-11-16
> **Regarding comparison with CANIFE**
>
> Now we would like to address your concern about the comparison to CANIFE. We absolutely concur that it is difficult to specify a good “ground truth” for comparison. We do not have any way to determine the “true” $\varepsilon$, even inefficiently, because doing so would require designing a provably optimal attack.
>
> But what reviewer ZyjW suggested is perhaps the best we can do: use a strong attack to determine a lower bound on $\varepsilon_\text{lo}$. If the lower bound exceeds $\varepsilon_{CANIFE}$, then we know CANIFE is underestimating $\varepsilon$. We still cannot rule out that our method is not overestimating $\varepsilon$, but at least it has not been invalidated by the lower bound.
>
> The CANIFE authors bill the method as a “measurement” of $\varepsilon$, not a lower bound. Nevertheless, CANIFE uses a lower bound on the per-round $\varepsilon_r$ to estimate the per-round $\sigma_r$. Thus, it has a built-in bias toward lower $\varepsilon$, and there is a maximum $\varepsilon$ it cannot exceed simply due to the finite sample size. CANIFE also searches for a canary example whose gradient is nearly orthogonal to a held-out set of data (*not* necessarily to the actual batch being audited). Our method uses a random gradient which is provably nearly orthogonal to the other gradients in the batch with high probability, leading to a stronger attack.
>
> So we ran the experiment reviewer ZyjW suggested. We inserted our random gradient canaries, thresholded on the cosine test statistics, and computed a $\varepsilon$ lower bound from the TPR and FPR of the optimal threshold. The result was a lower bound of $\varepsilon = 1.82 \pm 0.46$. This is indeed significantly higher than CANIFE’s $0.88 \pm 0.12$. It would be a stronger validation of our method if we could produce a lower bound that is closer to our value of $6.8 \pm 1.1$, but at least we can say that our estimate is in the plausible (but wide) range $(1.82, 34.5)$ while CANIFE’s is not. (34.5 is the analytical upper bound.) We will add this result and discussion to the CANIFE comparison in Appendix F.

---

> > ### Comment · Reviewer_zCeQ · 2023-11-18
> > **Thanks for the rebuttal**
> >
> > Thanks for addressing most of my concerns. When I wrote "I think for a setting in experiments you should also vary the dimensionality of the model and obtain a empirical privacy-dimensionality relationship (by keeping analytical one constant)." I meant to suggest increasing the model size of LSTM in Experiments section (for instance by increasing the number of weights in some layers) and reporting a similar result to table 2 with various dimensions; perhaps I was not clear enough. I know you don't have much time to do this experiments so I would suggest this as a future consideration.
> >
> > I'm leaning towards acceptance and will increase my score accordingly.

---

> > > ### Author Response · Authors · 2023-11-21
> > >
> > > Thank you for your comment.
> > >
> > > We apologize for the confusion about the experiment. We will try to include the experiment you suggested in the final revision of the paper.

---

### Official Review · Reviewer_LDHu · 2023-11-01

**Soundness:** 4 excellent
**Presentation:** 4 excellent
**Contribution:** 3 good
**Rating:** 8
**Confidence:** 3

**Summary:**

This paper proposes an approach for auditing the privacy of differentially-private learning algorithm in the context of federated learning such as DP-FedAvg. One of the benefits of the approach is that it is « one-shot » in the sense that it can be used at the same time as the model is learnt, without needing any retraining. It has also the advantage of being model agnostic.

**Strengths:**

The introduction clearly introduces the context of federated learning and motivates the need to develop empirical estimation methods to be able to audit the privacy provided by a differentially-private learning algorithm. The main challenges that need to be address to realize this are also clearly discussed. Overall, the paper is well-written and easy to follow.

The proposed approach has clear benefits over previous approaches, in the sense that it does not require retraining of the system or the use of well-crafted canaries. A comparison is also performed with CANIFE, which is one of the state-of-the-art method for privacy auditing. Overall, the obtained results demonstrate that the proposed method is promising in providing more tight privacy estimates.

**Weaknesses:**

The example analyzed in Table 1 assumes a high dimension as well as a large number of canaries, which is not particularly realistic. The authors should provide similar analysis for lower values of d and k. Similarly, in the experiments conducted the number of canaries used is quite large, which is likely to have an impact on the model utility. Thus, the authors should also reports the accuracy obtained for the model. In contrast, the values of epsilon used are very high and additional experiments should be performed with values of epsilon such as 0.1, 1, 5 and 10. Finally, experiments with a varying number of clients should also be conducted.

The difference between user-level and example-level differential privacy should be discussed and defined more clearly within the context of federated learning. For instance, how would the attack framework be impact by the change of definition, in particular with respect to the guarantees measured.

A small typo :
-« the choice gives » -> « this choice gives »

**Questions:**

It would be great if the authors could conduct additional experiments with a lower number of canaries as well as lower values of epsilon to be able to characterize how well the proposed method would fare in these situations.

---

> ### Author Response · Authors · 2023-11-13
>
> We appreciate your careful review and suggestions to improve our paper. We believe we can answer most of your concerns.
>
> * *In the experiments conducted the number of canaries used is quite large, which is likely to have an impact on the model utility.* We stated in the paper that for our Stackoverflow experiments, with 341k clients, “across the range of noise multipliers, the participation of 1k canaries had no significant impact on model accuracy – at most causing a 0.1% relative decrease”. (In fact, in an earlier draft we included a table comparing the accuracies, but since they are essentially unchanged, we thought the table was not helpful and wrote that sentence instead. To make this more clear to readers, we will add the table to the appendix.) There are many practical FL problems in which the number of real clients is high enough that adding 1000 canary clients will have negligible effect on model utility. For a real world benchmark, see Xu et al. (2023), which trains language models using DP-FL in a variety of languages, almost all of which have more than 3M clients each – ten times more than in our experiments.
> * *The example analyzed in Table 1 assumes a high dimension as well as a large number of canaries, which is not particularly realistic. The authors should provide similar analysis for lower values of d and k.* Many models of practical interest these days have more (often *far* more) than 1M parameters, including both of the benchmarks used in our experiments, and the industry language models cited above. We grant that our method would not be useful for small models, but 1M parameters is not very high by contemporary deep learning standards.
> * *The values of epsilon used are very high and additional experiments should be performed with values of epsilon such as 0.1, 1, 5 and 10.* One of the key points of our paper is that the analytical $\varepsilon$ vastly overstates the true privacy risk of releasing only the final model parameters. Notice that even at our smallest analytical $\varepsilon$ of 30, the empirical estimate $\varepsilon_\text{est}$-final is already less than 1. Look also at the density plots in the rightmost plot of Figure 1 corresponding to analytical $\varepsilon$ = 30, in which the canary cosine distributions are almost visually indistinguishable from the null hypothesis distribution. It is clear that adding even more noise to achieve lower values of analytical $\varepsilon$ would only lead to even more miniscule values of $\varepsilon_\text{est}$-all and $\varepsilon_\text{est}$-final, while destroying model utility. We would also point out that if the analytical $\varepsilon$ were as low as 1 or 0.1, we would have a theoretical guarantee of very strong privacy, so it would not be useful to employ our method to compute an empirical privacy estimate. Therefore we don’t think there is value in training models under such high levels of noise. We will add the statement of this reasoning to the text.
> * *The difference between user-level and example-level differential privacy should be discussed and defined more clearly within the context of federated learning.* We assume that DP-FedAvg would be used for user-level DP, and that DP-SGD would be used if only example-level DP is desired. It is conceivable that FedAvg could be modified somehow to achieve only example-level DP, but we feel that would be a strange thing to do, since user-level DP is the stronger form. When we note in the paper (footnote, page 1) that our approach can be modified to provide example-level DP, meaning that the key insight of using random nearly-mutually-orthogonal canaries carries over from DP-FedAvg to DP-SGD by using the canary vectors as example gradients instead of user updates. We will amend that footnote to be more explicit about the “trivial modification” required.
> * *Experiments with a varying number of clients should also be conducted.* As we argued above, there are many practical FL problems in which the number of real clients is high enough that adding 1000 canary clients will have negligible effect on model utility. We could run experiments with only a subset of the clients from our benchmark datasets, but the only result would be that the model utility would be lower.
>
> Have we answered all of your questions? Do you have any other concerns we could address to merit a higher score? Thank you.
>
> Z. Xu, Y. Zhang, G. Andrew, C. A. Choquette-Choo, P. Kairouz, H. Brendan McMahan, J. Rosenstock, Y. Zhang (2023). *Federated Learning of GBoard Language Models with Differential Privacy.* https://arxiv.org/abs/2305.18465.

---

> > ### Author Response · Authors · 2023-11-14
> > **Table 1 with different values of $d$**
> >
> > In order to address your concern, we ran the experiments on synthetic data reported in Table 1 with varying values of $d$. We set $k$ equal to $\sqrt{d}$ for each $d$. The results show that the method works well even down to $d$=10k, only it has higher variance. We will add the new values to the paper.
> >
> > Here we show, for each value of $d$, the mean and standard deviation of the $\varepsilon$ estimate for the three values of analytical $\varepsilon$: 1, 3, and 10.
> >
> > ```
> > d=10k
> > 0.98 +/- 0.41
> > 3.00 +/- 0.46
> > 9.89 +/- 0.71
> >
> > d=100k
> > 1.05 +/- 0.23
> > 3.00 +/- 0.31
> > 10.05 +/- 0.41
> >
> > d=1M
> > 0.99 +/- 0.14
> > 2.96 +/- 0.15
> > 10.00 +/- 0.23
> >
> > d=10M
> > 1.00 +/- 0.07
> > 3.00 +/- 0.08
> > 10.01 +/- 0.10
> > ```
> >
> > We would still stress however, that our method is intended for "normal scale" deep learning models with at least a million parameters or so.

---

> > > ### Comment · Reviewer_LDHu · 2023-11-22
> > >
> > > Thanks you very much for your answers and the experiments conducted, it address most of my concerns and I will increase my score accordingly.

---

### Author Response · Authors · 2023-11-18
**New revision uploaded**

As the discussion period comes to a close, we would like to gently request that reviewers LDHu and zCeQ confirm whether their questions have been answered.

We had a detailed discussion with reviewer ZyjW about our theoretical results, whose feedback helped us to make the proofs completely rigorous. We encourage the other reviewers to read that discussion.

We also ran an additional experiment requested by reviewer ZyjW to provide a better “ground truth” for our comparison with CANIFE. We believe this also addresses one of the concerns of reviewer zCeQ that “it is hard to interpret the provided empirical comparison with CANIFE”.

We have uploaded a new revision of the submission with these changes. We have not had time to incorporate all of the reviewers' minor suggestions in this revision, but we will make sure to do so for the camery ready version.

If there is any other question or concern that we could address in order to merit a higher score, please let us know. Thank you for taking time to review and help us improve our submission.

---

### Meta-Review · Area_Chair_qQhh · 2023-12-05

**Metareview:**

The paper presents a method for privacy auditing high-dimensional federated learning in the single training run.

Strengths: important problem, carefully presented, elegant and practical method, promising empirical results.

Weaknesses: no serious weaknesses as far as I can tell. Maybe a slight concern is that the method would seem to estimate an average epsilon, not worst case like DP. This may cause the method to fail to detect some DP violations that do not affect all data uniformly. Adding some warning on this to the Conclusion might be a good idea.

**Justification For Why Not Higher Score:**

N/A

**Justification For Why Not Lower Score:**

I believe this is a very strong paper, presenting a clean, elegant and practical method for the important problem of privacy auditing in DP federated learning.

---

### Decision · Program_Chairs · 2024-01-16

Accept (oral)